# Phagocytosis is mediated by two-dimensional assemblies of the F-BAR protein GAS7

Kyoko Hanawa-Suetsugu[1,10], Yuzuru Itoh[2,8,10], Maisarah Ab Fatah[1,10], Tamako Nishimura[1,10], Kazuhiro Takemura[3], Kohei Takeshita[4], Satoru Kubota[1], Naoyuki Miyazaki[5], Wan Nurul Izzati Wan Mohamad Noor[1], Takehiko Inaba[1], Nhung Thi Hong Nguyen[1], Sayaka Hamada-Nakahara[2], Kayoko Oono-Yakura[1], Masashi Tachikawa[6], Kenji Iwasaki [5,9], Daisuke Kohda [4,7], Masaki Yamamoto[4], Akio Kitao [3], Atsushi Shimada [4,7]★ & Shiro Suetsugu [1]★

Phagocytosis is a cellular process for internalization of micron-sized large particles including pathogens. The Bin-Amphiphysin-Rvs167 (BAR) domain proteins, including the FCH-BAR (F-BAR) domain proteins, impose specific morphologies on lipid membranes. Most BAR domain proteins are thought to form membrane invaginations or protrusions by assembling into helical submicron-diameter filaments, such as on clathrin-coated pits, caveolae, and filopodia. However, the mechanism by which BAR domain proteins assemble into micron-scale phagocytic cups was unclear. Here, we show that the two-dimensional sheet-like assembly of Growth Arrest-Specific 7 (GAS7) plays a critical role in phagocytic cup formation in macrophages. GAS7 has the F-BAR domain that possesses unique hydrophilic loops for two-dimensional sheet formation on flat membranes. Super-resolution microscopy reveals the similar assemblies of GAS7 on phagocytic cups and liposomes. The mutations of the loops abolishes both the membrane localization of GAS7 and phagocytosis. Thus, the sheet-like assembly of GAS7 plays a significant role in phagocytosis.

[1] Nara Institute of Science and Technology, Ikoma 630-0192, Japan. [2] University of Tokyo, Tokyo 113-0032, Japan. [3] School of Life Science and Technology, Tokyo Institute of Technology, Tokyo 152-8550, Japan. [4] RIKEN SPring-8 Center, Sayo, Hyogo 679-5148, Japan. [5] Institute for Protein Research, Osaka University, Suita, Osaka 565-0871, Japan. [6] Theoretical Biology Laboratory, RIKEN, Wako 351-0198, Japan. [7] Division of Structural Biology, Medical Institute of Bioregulation, Kyushu University, Fukuoka 812-8582, Japan. [8]Present address: Science for Life Laboratory, Department of Biochemistry and Biophysics, Stockholm University, Stockholm, Sweden. [9]Present address: Tsukuba Advanced Research Alliance, Life Science Center for Survival Dynamics, University of Tsukuba, Tsukuba, Japan. [10]These authors contributed equally: Kyoko Hanawa-Suetsugu, Yuzuru Itoh, Maisarah Ab Fatah, Tamako Nishimura. *email: ashimada@biureg.kyushu-u.ac.jp; suetsugu@bs.naist.jp

Phagocytosis is a process that internalizes larger, micron-sized particles than those internalized by other endocytic pathways, such as clathrin-mediated endocytosis[1–3]. The plasma membrane extends and engulfs phagocytic particles. The proteins of the BAR domains have a more rigid membrane-binding surface than the membrane itself, and the architecture of this surface, that is, concave or convex, reflects the mechanisms for membrane curvature generation[4]. The BAR domain proteins with concave membrane-binding surfaces are classified as the (N-)BAR and F-BAR domain subfamilies and function in plasma membrane invaginations with submicron diameters, including clathrin-coated pits and caveolae[5–8]. In contrast, the I-BAR domains with convex surfaces, such as that of IRSp53, function to create protrusions including filopodia[9]. Most BAR, F-BAR and I-BAR domains are thought to form membrane invaginations or protrusions by assembling into submicron-diameter filaments[10–12]. However, the means by which the BAR domain superfamily proteins assemble into micron-scale structures on phagocytic cups has remained unclear.

GAS7 is expressed in the brain, spleen, lung, testis[13–15] and immune cells, including macrophages, which are capable of phagocytosis[16,17]. GAS7 regulates the formation of membrane protrusions in neurons[13,18] and invasive lung cancer cells[14]. Some receptors involved in phagocytosis are co-expressed with GAS7 according to Coxpres DB, a database of gene-expression correlations[19]. However, thus far, the function of GAS7 in macrophages, cells highly capable of phagocytosis, has not been studied.

In this study, we show that GAS7 plays a critical role in phagocytic cup formation in macrophages. Crystallographic, electron microscopic, biochemical and cellular localization analyses revealed that the GAS7 F-BAR domain possesses unique hydrophilic loops that contribute to two-dimensional sheet formation of GAS7 on flat membranes of phagocytotic cup, indicating that the sheet-like assembly of GAS7 is essential for phagocytosis.

## Results

**GAS7 splicing isoforms.** To assess the possible involvement of GAS7 in phagocytosis, we examined the isoforms of GAS7 expressed in macrophages by comparing the molecular weights of various expressed, non-tagged GAS7 isoforms in HeLa cells to that of the endogenous GAS7 in RAW264.7 macrophages. GAS7b (*Mus musculus* and *Homo sapiens*), GAS7cb (*Mus musculus*), and GAS7c (*Homo sapiens*) are GAS7 splicing isoforms (Supplementary Fig. 1a). In addition to the F-BAR domain, GAS7b possess a WW domain, and GAS7cb and GAS7c possesses a WW and an SH3 domain, while GAS7d contains only the F-BAR domain. As determined by western blotting, HeLa cells do not express endogenous GAS7, while GAS7b is highly expressed in RAW264.7 macrophages (Supplementary Fig. 1b, c).

**Two-dimensional assembly of GAS7 on the membrane in vitro.** The F-BAR domains often bind to negatively charged membranes, where the negative charge is provided by phosphatidylserine (PS) and phosphoinositides, including phosphatidylinositol (3,4,5)-trisphosphate ($PIP_3$), which are enriched in the phagocytic cups[20]. To examine the membrane binding of GAS7, giant unilamellar vesicle (GUV) liposomes containing PS and $PIP_3$ were incubated with GAS7 tagged with green fluorescent protein (GFP), and GAS7 assembly was observed by fluorescence microscopy (Fig. 1a). The isolated F-BAR domain and GAS7b both assembled on GUVs without prominent membrane deformation or tubulation (Fig. 1a). Occasionally, the F-BAR domain and GAS7b partially covered the surfaces of GUVs. The amount of GAS7 on the partially covered surface increased in a time-dependent manner, suggesting that the GAS7 assembly continued on the GAS7-bound membrane

(Fig. 1b). Interestingly, the regions of GUVs without GAS7 were later completely covered by GAS7, with the progression at the edges of the GAS7 assembly. Therefore, these results suggested that the binding of GAS7 to the membrane occurs at the pre-existing assemblies of GAS7 on the membrane.

To understand GAS7 assembly on the membrane in detail, we prepared flat membranes by forming monolayered phospholipids on the grid for electron microscopic observations[21,22]. We observed GAS7 on lipid monolayers by electron microscopy (Fig. 1c). The monolayers showed the striations formed by the GAS7 F-BAR domain, GAS7b and GAS7cb, all of which indicated the sheet-like GAS7 assemblies with striations similar to each other on the membrane (Fig. 1c). Consistent with the possible multiple assembly sites of GAS7 on GUVs, there were multiple striations of GAS7 in random directions, suggesting that there are large numbers of GAS7 assemblies on the membrane. The apparent pitches of striation were ~5 nm, as estimated by the Fourier transform of the images (Fig. 1d, e). These striations were not resulted from the membrane because the striation was not observed without protein (Fig. 1c), suggesting the higher order assembly of GAS7 by its F-BAR domain, as shown for FBP17 and CIP4 on the flat membrane[10].

**Structure of the GAS7 F-BAR domain.** To understand the functions of GAS7, we determined the crystal structure of the GAS7 F-BAR domain, using the crystals of an F-BAR fragment and GAS7cb (Fig. 2a, Supplementary Fig. 1d, Supplementary Table 1). In the crystals of GAS7cb, only the F-BAR domain was visible in the electron density maps. The structures of the F-BAR domains in both crystals were almost identical, with a root mean square deviation of 1.2 Å. The GAS7 F-BAR domain forms a helical-bundle dimer with a shallow concave curvature, which is one of the flattest curvatures amongst the known F-BAR domains (Supplementary Fig. 2a, b).

The asymmetric unit of the F-BAR domain crystal contained two F-BAR dimers. These dimers interacted with their symmetry-related dimers, forming filamentous oligomers with asymmetric flat surfaces, which we refer to as flat filamentous oligomers (FFOs) (Fig. 2b). The two FFOs formed by the different dimers in the asymmetric unit were essentially identical to each other. The FFOs had a width of ~14 nm with a ~5 nm distance between adjacent F-BAR dimers, while the F-BAR dimers were ~22 nm in length. The lateral alignment of F-BAR domains in FFO appeared to be similar to the pitch of the GAS7 striations on the monolayered membrane (Fig. 1d, e). In the GAS7cb crystals, the periodicity observed on the monolayered membrane appeared to be absent (Supplementary Fig. 2c). Therefore, the GAS7 striations on the membrane might be formed by the assembly of the FFOs.

In CIP4 and FBP17, the contacts between the F-BAR domains in the crystals were consistent with those on the membrane[23]. We hypothesized that the GAS7 filament in the crystal could have physiological relevance, including membrane binding. Therefore, we considered the possible positions of the WW and SH3 domains of GAS7 on this filament. The WW and SH3 domains might be positioned on one side of the FFO, because there are huge spaces between the domains for the linker region between the F-BAR and the WW domain to pass through, as shown hypothetically in Fig. 2c.

**FFL2 interacts with the membrane without insertion.** The striations on the monolayered membrane suggested the possible formation of FFOs, as in the crystal. The FFOs of the F-BAR domain in the crystals appeared to be mediated by two GAS7-specific loops, filament forming loop 1 (FFL1) (aa 171–197; GAS7cb amino-acid residue numbers) and filament forming loop

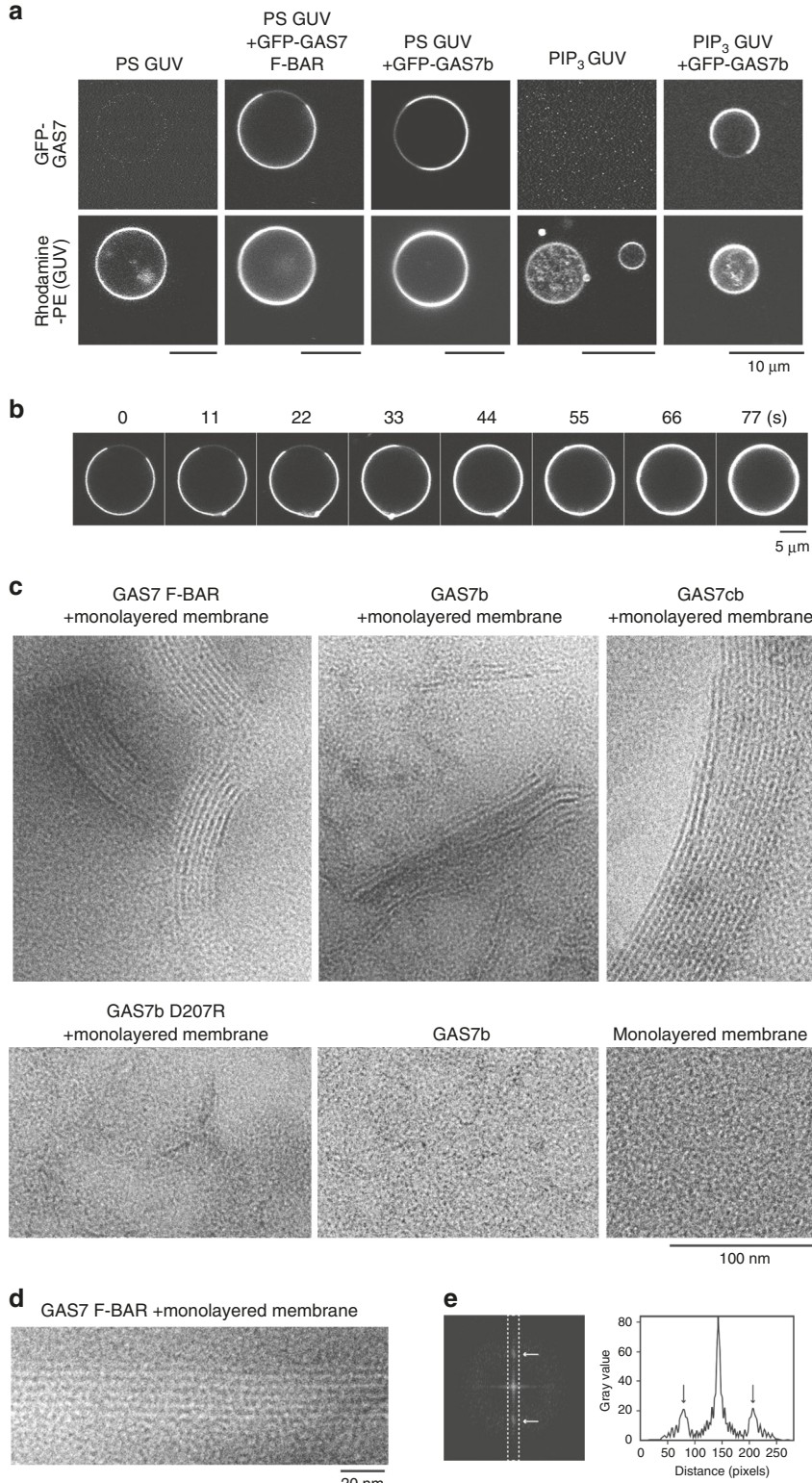

2 (FFL2) (aa 206–219), which are located on the convex and concave surfaces, respectively (Fig. 2a, d, Supplementary Fig. 1d). The positions of the two FFL1s and FFL2s are symmetrical; however, one of the two FFL1s at the convex surface of the dimer interacted with the tip of the adjacent dimer, and one of the two FFL2s at the concave surface of the dimer interacted with a conserved region on the convex surface of the adjacent dimer in the FFO, thus tilting the helix bundle relative to the plane of the

FFO (Fig. 2d, Supplementary Fig. 2d–f). Due to asymmetry, one side of the flat surfaces of the FFO is more positively charged than the other (Fig. 2e). Thus, we supposed that this surface could be the membrane-binding surface. These contacts in the FFO were unique and did not resemble any interactions used for oligomer formation by other BAR, I-BAR and F-BAR domains, including the Pinkbar I-BAR domain, which also forms a planar oligomer (Supplementary Fig. 2g) localized close to the cell–cell contacts[24].

**Fig. 1** Membrane binding by GAS7 as the flat two-dimensional sheet. **a** Giant unilamellar vesicles (GUVs) were incubated with 0.5 μM GFP-GAS7 F-BAR domain fragment or GAS7b. The proteins were incubated with GUVs containing PC, PE, PS and rhodamine-PE at a molar ratio of 20:20:60:0.02, and GUVs containing PC, PE, PS and PIP$_3$ at a molar ratio of 40:40:20:5 were observed at 37 °C. Scale bar: 10 μm. **b** The time course of GFP-GAS7b assembly on GUVs. Scale bar: 5 μm. **c** Negatively stained transmission electron micrographs of the GAS7 F-BAR domain fragment, GAS7b, GAS7cb and GAS7b D207R mutant on the monolayered membrane. Negatively stained transmission electron micrograph of GAS7b alone and the membrane alone are also shown. Protein samples (0.1 μM) were incubated in the presence or absence of lipid monolayers containing PC, PE and PS at a molar ratio of 20:20:60. Scale bar: 100 nm. **d** Negatively stained transmission electron micrograph of the GAS7 F-BAR domain fragment in the monolayered membrane as in **c** for **e**. Scale bar: 20 nm. **e** The Fourier transform of the micrograph in **d**. Arrows indicate signals in the reciprocal space corresponding to the periodic striations in **d** with the periodic spacing of ~5 nm. A cross-section of the diffraction in the Fourier image is also shown. Arrows indicate the signals showing the regular spacing of ~5 nm

One of the two FFL2s lacks contacts with the adjacent dimer and protrudes from the FFO surface (Supplementary Fig. 2a), suggesting the direct contact of the FFL2 with the membrane. Moreover, the FFL2s are mostly composed of hydrophilic residues (Supplementary Fig. 1d). The membrane binding of GAS7 was then experimentally examined using GAS7b, the isoform highly expressed in macrophages, by a liposome co-sedimentation assay. Referring to the other BAR domain studies, we used liposomes reconstituted from bovine brain Folch fraction lipids[23,25,26], which contained a high amount of PS. GAS7b bound to the PS and PIP$_3$ containing reconstituted liposomes equally well, confirming the relatively low sensitivity to the negatively charged phospholipid species, as also shown by GUV-binding experiments (Fig. 2f, Supplementary Fig. 4a). The replacement of Gln212 at the tip of FFL2 with Arg (Q212R) did not alter the membrane interaction under the physiological salt conditions, but strengthened the membrane binding at a modestly higher salt concentration that reduced the electrostatic interaction (Fig. 2g, h, Supplementary Fig. 4b, c). Given that the introduction of Arg to the tip of FFL2 promoted the membrane binding, it appeared to be unlikely that this loop is inserted into the membrane, as in the case of the 'wedge loop' of the F-BAR domains of PACSINs[27,28], which contains more hydrophobic residues, especially at the tip of the loop.

To examine the importance of FFL2, we analysed its deletion mutants. The binding of GAS7b ΔFFL2 mutants to liposomes consisting of the brain Folch fraction was weaker than that of GAS7b (Fig. 2k, Supplementary Fig. 4d). There are several positively charged amino-acid residues on FFL2 (Supplementary Fig. 1d). The replacement of Lys209 with Glu and that of Lys208 and Lys209 with Ala both resulted in reduced binding to liposomes consisting of brain Folch lipids and to reconstituted liposomes containing PS (Fig. 2i, j, Supplementary Fig. 4e, f). These data indicated that the positively charged residues of FFL2 play a crucial role in the membrane binding of GAS7 through electrostatic interactions, again supporting the non-insertion of FFL2 into the membrane.

The above mutants indicated the importance of FFL2 for the membrane binding of GAS7; however, the role of FFL2 in FFO formation was not addressed. The FFLs and tips are the most flexible parts of the F-BAR dimer, as shown by the molecular dynamics (MD) simulation, although their configurations relative to the α-helix bundle did not change during the simulation (Supplementary Fig. 3a, b). Asn177 of FFL1 and Asp207 of FFL2 were in contact with Arg326 on helix α2b (Fig. 2a, Supplementary Fig. 3c), suggesting that these residues stabilized the configuration of the FFLs. The replacement of Asp207 with Arg, which increases the positive charge, was expected to increase the membrane binding simultaneously with the FFL2 destabilization by removing its interaction with the α-helix bundle. Interestingly, GAS7b and the D207R mutant exhibited similar binding to reconstituted liposomes containing PS, as examined by a co-sedimentation assay (Fig. 2l, Supplementary Fig. 4g). To examine

the oligomer formation of GAS7, we treated GAS7b with chemical cross-linkers. Highly cross-linked GAS7b was observed after the treatment with chemical cross-linkers as a mobility shift in electrophoresis, which suggested the trimer or tetramer formation by GAS7, although the larger oligomers of GAS7 shown in the FFO model were not easily resolved by electrophoresis (Fig. 2m). In the presence of the liposomes, the increase in the oligomer bands of GAS7b were observed, suggesting the promotion of oligomer formation on the membrane. The D207R mutant exhibited fewer oligomer bands than GAS7b in the electrophoresis after the chemical cross-linking reaction in the presence of the liposomes, suggesting that the oligomerization of the D207R mutant was different from that of GAS7b (Fig. 2m, Supplementary Fig. 4h). Consistently, the striations of D207R mutant on the membrane were different and reduced from those of GAS7b (Fig. 1c). Because the increase of arginine would strengthen the binding to the membrane, the D207R mutation was thought to compensate for the reduction in the oligomerization that would contribute to the membrane binding, suggesting the reason of the membrane binding of the D207R mutant similar to that of GAS7b. Therefore, the GAS7 oligomer formation on the membrane appeared to be promoted over that in the solution, where the configuration of FFL2 was suggested to be important for the adequate oligomerization of GAS7.

**The membrane-binding surface of the GAS7 F-BAR domain.** The N-surface is perpendicular to the dimer axes, and corresponds to the concave surfaces of the BAR and F-BAR domains and the convex surface of the I-BAR domain (Fig. 2b)[29]. The N-surface in these domains often contains more conserved residues than those in the other surfaces, and is the membrane-binding surface for most BAR, F-BAR and I-BAR domains[23]. However, the concave N-surface of the GAS7 F-BAR domain dimer is less conserved than the other surfaces (Supplementary Fig. 2d). This is in striking contrast to the other F-BAR domains, such as CIP4 and FBP17, with membrane-binding surfaces containing numerous highly conserved residues[23]. Thus, the N-surface may not be the membrane-binding surface of GAS7, in agreement with the asymmetry of the F-BAR in the FFO model (Fig. 2b).

FFL1 is thought to be distant from the membrane, when membrane binding through the N-surface is assumed. Therefore, we examined the membrane binding of the deletion mutant of FFL1. The binding of the GAS7b ΔFFL1 mutant to liposomes consisting of brain Folch lipids was weaker than that of GAS7b (Fig. 2g, Supplementary Fig. 4b). This affinity reduction suggested that interaction sites other than the N-surface are involved in the membrane binding of GAS7. Indeed, if we assume the FFO model, then this result is reasonable, as FFL1 is thought to be a critical interaction site for oligomer formation.

The membrane binding of GAS7 is likely to be mediated by electrostatic interactions. Therefore, the positively charged

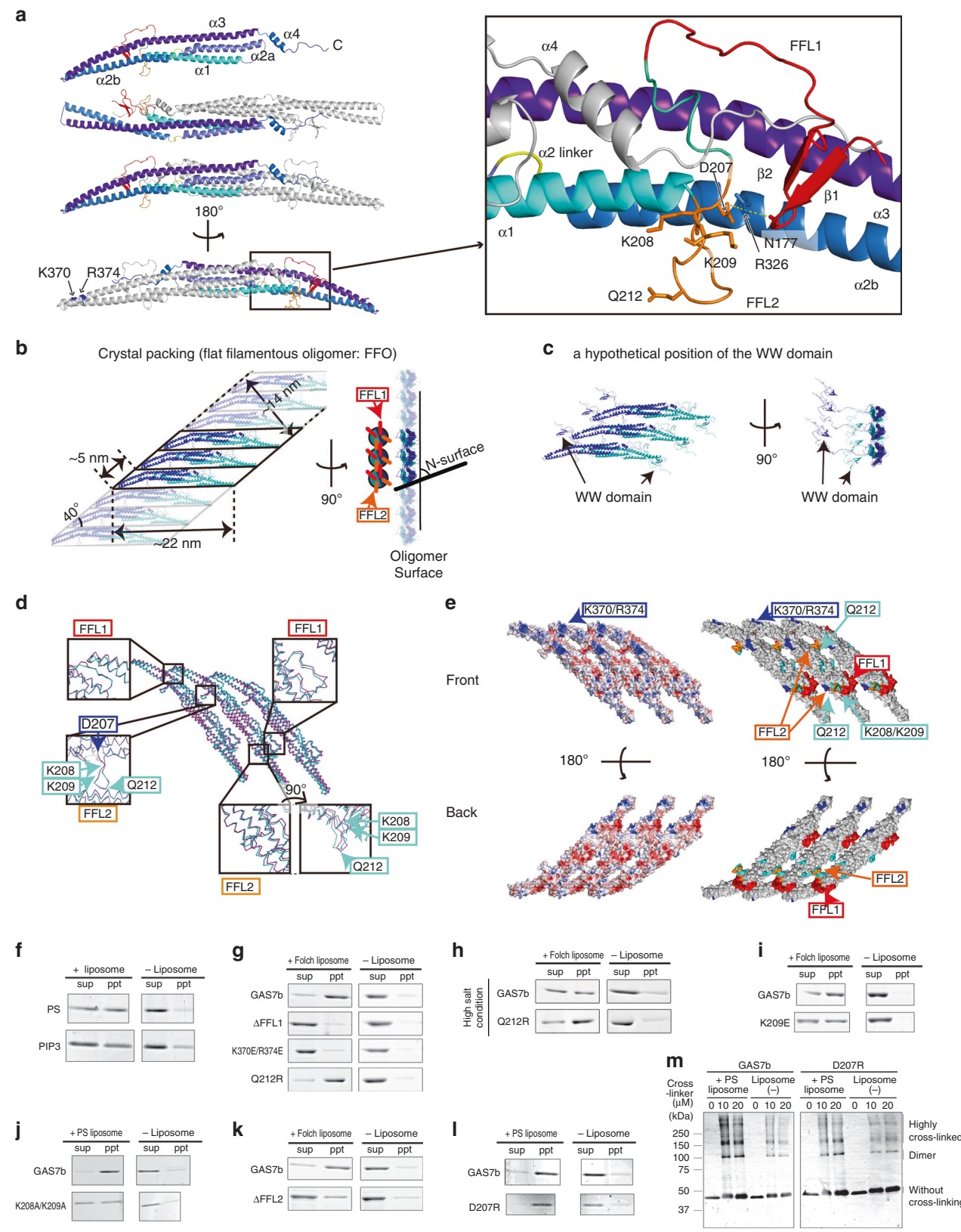

amino-acid residues, Lys370 and Arg374, on the side surface of the dimer, that is, the front surface of the FFO, were examined (Fig. 2a). When these residues were both replaced with glutamate, membrane binding was abolished (Fig. 2g, Supplementary Fig. 4b), further suggesting the membrane binding of GAS7 through the FFO surface.

The assembly and the membrane localization of GAS7 were examined in HeLa cells, which do not express any isoforms of endogenous GAS7 (Supplementary Fig. 1b) and therefore could be a good model system to examine the assembly of GAS7 within cells. When GAS7b and the F-BAR domain fragment of GAS7 were overexpressed in HeLa cells, both GAS7b and F-BAR

**Fig. 2** Crystal structures of GAS7 and the membrane-binding sites. **a** Ribbon diagrams of the crystal structure of the isolated GAS7 F-BAR domain and the GAS7-specific N-terminal region including FFL1 (red) and FFL2 (orange). Secondary structural elements are indicated. R326 on helix α2b contacts N177 on FFL1 as well as D207 on FFL2. The K208, K209, Q212, K370 and R374 residues are also indicated. **b** Crystal lattice of the isolated F-BAR domain. The asymmetric unit of the crystal contains two F-BAR dimers, which both form the FFO by the crystal packing. The N-surface, which is perpendicular to the dimer axes, is indicated with the illustration of FFL1 and FFL2. **c** The hypothetical position of the WW domain of GAS7b on the FFO. The two WW domains of the F-BAR dimer can be localized to one side of the FFO. **d** Superimposition of two FFOs composed of different F-BAR dimers in the asymmetric unit of the crystal (cyan and magenta). The contact sites between the F-BAR domains are indicated. **e** Surfaces of the FFO. (left) Electrostatic surface potentials of FFOs composed of three F-BAR dimers; blue and red indicate positive and negative charges, respectively. (Right) In the same orientation as in **d**. The point mutations with defects in the F-BAR domain assembly (blue), and those without defects (cyan) are shown according to Supplementary Fig. 5. FFL1 (red), FFL2 (orange), Q212 (cyan), K208/K209 (cyan) and K370/R374 (blue) are also indicated. **f–l** GAS7b and its ΔFFL1, K370E/R374E, Q212R, ΔFFL2, K209E, K208A/K209A and D207R mutants were examined for their binding to the liposomes by liposome co-sedimentation assays. The presence of proteins in the pellet (ppt) indicates membrane binding. sup: supernatant. The liposomes were of the bovine Folch fraction (**g–i**, **k**), the PC, PE and PS lipids at a ratio of 20, 20 and 60 (**f**, **j**, **l**), and the PC, PE, PS and PIP$_3$ lipids at a ratio of 40, 40, 20 and 5 (**f**). **m** Cross-linking of GAS7b and the D207R mutant treated with the BS(PEG)5 (PEGylated bis(sulfosuccinimidyl)suberate) cross-linker, in the presence or absence of PS liposomes

assembled into patches with cup-like structures of several microns (Fig. 3a, Supplementary Fig. 5a). The GAS7 localization on the membrane was apparently different from those of other plasma membrane-localized BAR domains with the ability to bind to relatively flat membranes, such as Pinkbar I-BAR and the Nwk ortholog FCHSD1 F-BAR[24,30] (Supplementary Fig. 5b). Interestingly, the patches excluded the palmitoylated DsRed-monomer (DsRed-membrane), a diffusive plasma membrane marker, suggesting that the patches were sheets composed of highly concentrated GAS7b and GAS7 F-BAR (Fig. 3a, Supplementary Fig. 5a). The patch formation was abolished by the membrane-binding deficient ΔFFL1, ΔFFL2 and K370E/R374E mutations, whereas the membrane-binding Q212R mutant formed patches (Fig. 3a, Supplementary Fig. 5a). The D207R mutation abolished patch formation (Fig. 3a), which was thought to be consistent with the altered oligomer formation in vitro (Fig. 2m). Thus, the defect in the FFL2-mediated oligomer formation is thought to be essential for patch formation.

We also mutated the other positively charged residues. The K312E/K313E and K316E/K317E mutants did not exhibit the assembly, in contrast to the K279E/K280E and K449E/K450E mutants (Supplementary Fig. 5a). The effective mutations are located between the side and the N-surface of the F-BAR domain dimer structure, suggesting that the FFO surface or a similar one is the membrane-binding surface (Fig. 2e, Supplementary Fig. 5a).

**GAS7b assembly on the plasma membrane for phagocytosis.** In HeLa cells, the GAS7 F-BAR domain fragment patches sometimes showed invaginations with micro-size diameters, suggesting the possible membrane deformation by the F-BAR domain (Supplementary Fig. 5a). When GAS7b was expressed and observed in live cells, the GAS7b patches appeared at ruffled membranes (Fig. 3b) and eventually transformed into holes reminiscent of phagocytosis (Fig. 3c). Therefore, we examined the localization of GAS7b in RAW264.7 macrophages. An incubation with zymosan, a phagocytic substrate derived from a yeast protein–carbohydrate complex, induced phagocytic cup formation with localized GFP-tagged GAS7b (Fig. 3d), which was expressed in GAS7-knockout RAW264.7 macrophages at a similar level to the endogenous GAS7b (Supplementary Fig. 6). Immunofluorescence microscopy revealed the accumulation of endogenous GAS7b with actin filaments at phagocytic cups surrounding the zymosan particles (Fig. 3e). N-WASP and Arp3, a binding partner of GAS7 for actin polymerization and a subunit of the Arp2/3 complex responsible for the N-WASP-mediated actin polymerization[13,14], colocalized with GAS7b (Fig. 3f, g). Receptors for phagocytosis, such as complement receptor CR3 component CD11b[31] and mannose receptor CD206[32], also colocalized with GAS7b at phagocytic cups (Supplementary Fig. 7a, b).

The phagocytic activity of the RAW264.7 macrophage cells with the reduced expression or knockout of GAS7 was examined, using zymosan. The zymosan uptake was reduced in both GAS7 small interfering RNA (siRNA)-treated and GAS7-knockout cells (Supplementary Fig. 7c–e). IgG-coated bead uptake was also reduced in GAS7-knockout cells (Supplementary Fig. 7f). Consistent with the localization of GAS7 at lamellipodia, the GAS7-knockout cells were defective in lamellipodia formation (Supplementary Fig. 7g, h). The zymosan phagocytosis was rescued by the endogenous-level forced expression of GAS7b and GAS7cb in the knockout cells, but not by the GAS7b ΔFFL2 mutant or the F-BAR domain fragments (Fig. 3h, i, Supplementary Fig. 7i). Neither the D207R nor K209E mutant restored phagocytosis, indicating that the oligomerization and the membrane binding of GAS7 were essential for the phagocytic cup formation (Fig. 3h, j).

**Similar GAS7b assemblies on membrane in vitro and in cells.** The assemblies of GAS7b on GUVs, in HeLa cells, and in macrophage cells were compared by fluorescence recovery after photobleaching (FRAP). On the GUVs, both the F-BAR domain fragment and GAS7b exhibited slow turnovers, although GAS7b exhibited slightly faster recovery after photobleaching (Fig. 3k, Supplementary Fig. 8a, b). Similarly, the GAS7b within a patch in HeLa cells exhibited slower turnover than the GAS7b in the cytosol, but the turnover of the F-BAR domain fragment was slower than that of the GAS7b in a patch (Fig. 3l, Supplementary Fig. 8d–f). Interestingly, the GAS7b in the phagocytic cups of macrophages exhibited a similar turnover to that in the GAS7b patches formed in HeLa cells, suggesting similar GAS7b assemblies in these cells (Fig. 3l, Supplementary Fig. 8c, d).

Next, the assembly of GAS7b at the single-molecule resolution on membranes was visualized and compared with a simulated GAS7b localization (Fig. 4a). The localization was determined using GAS7b tagged with mEOS4b, a photoconvertible fluorescent protein that enables stochastic observations for super-resolution imaging[33]. The mEOS4b-GAS7b bound to GUVs (Fig. 4b, c) became localized to phagocytic cups, and rescued phagocytosis in GAS7-knockout cells (Fig. 4d, e, Supplementary Fig. 7i). GAS7b assembled into sheets, possibly by the alignment of the FFOs on the lipid monolayer (Fig. 1c). Thus, we created a model of the GAS7b sheet that resembles the electron microscopic observations of the lateral alignment of the FFOs (Fig. 4a). The assembly of mEOS4b-GAS7b was then examined by comparing the occurrence of neighbouring molecules, with random localizations and simulated FFO sheets (Fig. 4a). The occurrences of the observed neighbours of GAS7b on liposomes and on the top of phagocytic cups were similar to each other, and more frequent than those of the non-specifically adsorbed GAS7b on the glass and the bottoms of the phagocytic cups (Fig. 4f–i). Furthermore, the GAS7b

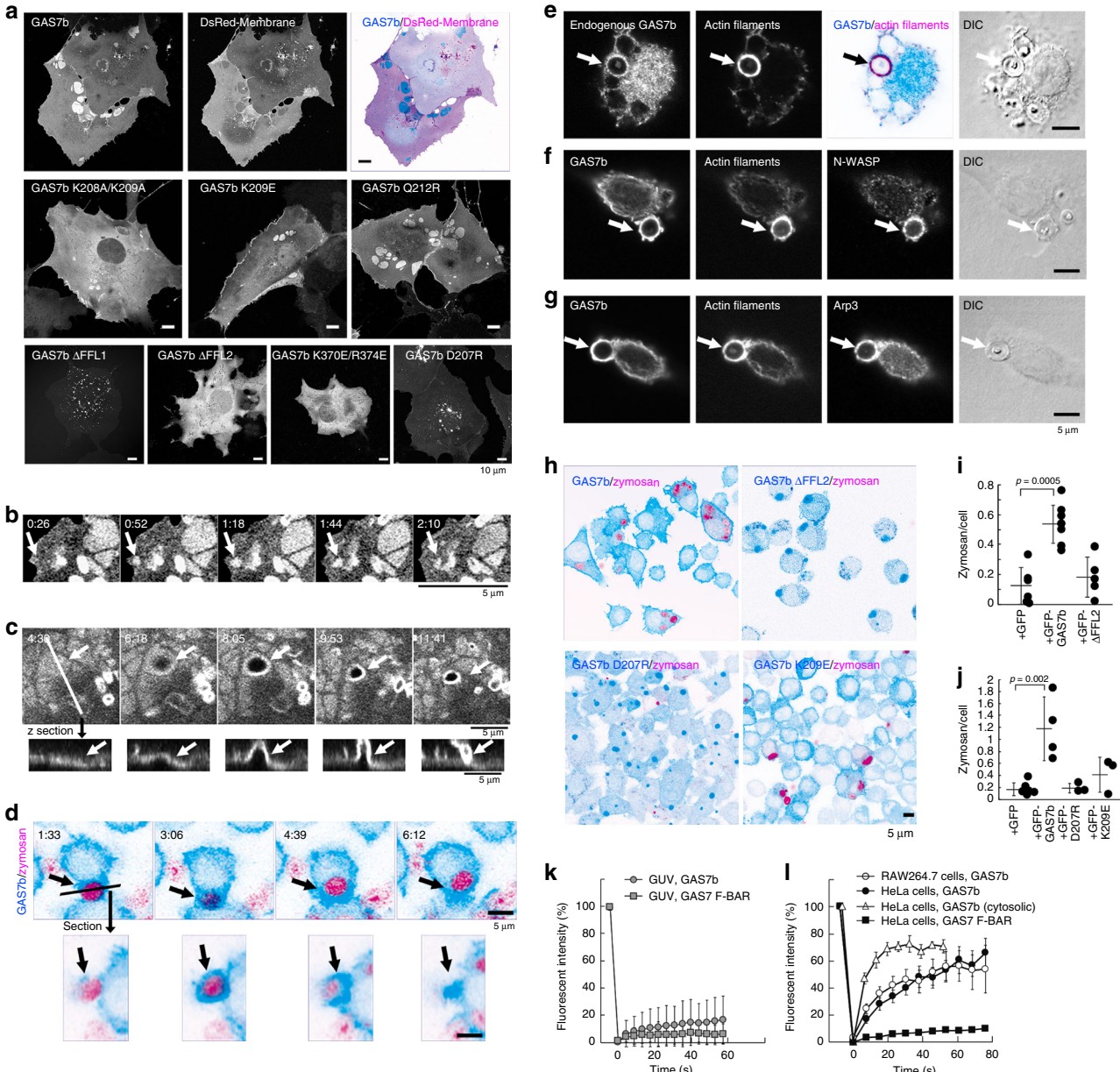

**Fig. 3** GAS7b sheet formation in HeLa cells and macrophages. **a** GFP-GAS7b and its ΔFFL1, ΔFFL2, D207R, K208A/K209A, K209E, Q212R and K370E/R374E mutants expressed in HeLa cells. Plasma membranes of the cells were visualized by the co-expression of palmitoylated DsRed (DsRed-membrane: magenta) with GAS7b (cyan), to highlight the dense assembly that excludes the DsRed-membrane. Scale bars: 10 μm. **b**, **c** Time-lapse images of HeLa cells expressing GFP-GAS7b. Arrows indicate the appearance of the sheet-like GAS7b patches at ruffles (**b**) and the sheet-like GAS7b patches that transformed into holes as shown in the z sections (**c**). Scale bars: 5 μm. **d** Time-lapse images of GFP-GAS7b (cyan) expressed in GAS7-knockout RAW264.7 macrophages incorporating the phagocytosis substrate, zymosan (magenta), captured at 93 s intervals. Arrows indicate the zymosan incorporation. The section at the line is also shown. Scale bar: 5 μm. **e** Localizations of endogenous GAS7 (cyan) and actin filaments (magenta) in RAW264.7 macrophages incorporating zymosan (DIC, arrows). Scale bar: 5 μm. **f**, **g** N-WASP (**f**) and Arp3 (**g**) with GFP-GAS7b (cyan) stably expressed at the endogenous GAS7 level in GAS7-knockout RAW264.7 macrophages incorporating zymosan (DIC, arrow). Scale bar: 5 μm. **h** Localizations of GFP-GAS7b, ΔFFL2, D207R and K209E mutants (cyan) in GAS7b-knockout RAW264.7 macrophages after the incorporation of zymosan (magenta). Scale bar: 5 μm. **i**, **j** Quantification of zymosan incorporation by GAS7b-knockout RAW264.7 macrophages stably expressing GFP, GFP-GAS7b and ΔFFL2 mutant (**i**) or GFP, GFP-GAS7b, GFP-GAS7b, D207R mutant and K209E mutant (**j**). Dots represent zymosan incorporation by each cloned cell line. GFP alone was expressed as a control. *P* values were determined by the two-tailed Student's *t* test relative to GFP-expressing GAS7-knockout cells are shown. Error bars show SD. Source data are provided as a Source Data file. **k**, **l** Time courses of fluorescence recovery after photobleaching for GFP-GAS7 F-BAR and GFP-GAS7b on GUVs prepared from the PC, PE and PS lipids at a ratio of 20:20:60 (**k**), and GFP-GAS7 F-BAR and GFP-GAS7b expressed in HeLa cells and GFP-GAS7b expressed at the endogenous level in GAS7-knockout RAW264.7 macrophages (**l**). Source data are provided as a Source Data file. Error bars show SD

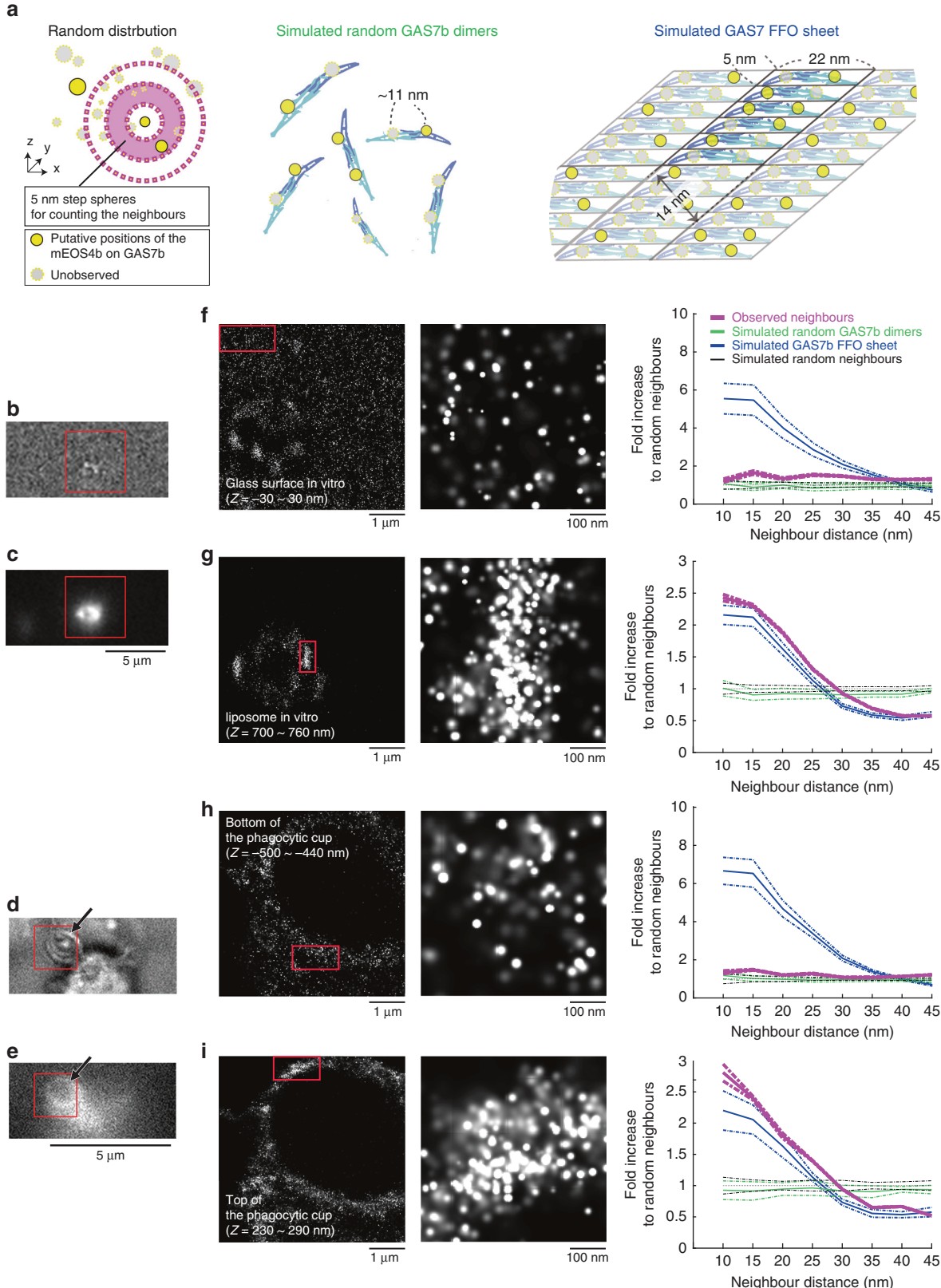

observed on liposomes and the tops of phagocytic cups had neighbours occurring with similar frequencies to those of the simulated FFO sheets (Fig. 4g, i). Interestingly, these GAS7b signals were sometimes aligned adjacent to each other (Fig. 4g, i). These data support the proposal that the GAS7 assembly into sheets occurs on both liposomes and phagocytic cups.

## Discussion

Most BAR domain superfamily proteins are involved in the generation of membrane structures with submicron diameters, including clathrin-coated pits, caveolae, filopodia, endosomes and so on, through their membrane-binding abilities[6,34]. Several BAR superfamily proteins are involved in forming and binding to

**Fig. 4** GAS7b sheets on liposomes and phagocytic cups of macrophages. **a** Illustration of the possible spatial distributions of super-resolution signals. Randomly distributed signals, signals from randomly placed GAS7b dimers and signals from GAS7b forming FFO sheets are schematically illustrated. The alignment of the FFOs in the simulated sheet was considered to be lateral, as illustrated. Dashed magenta circles indicate the counting of the signals neighbouring a signal. **b–e** Phase-contrast (**b**, **d**) and epi-fluorescent (**c**, **e**) images of liposomes (**b**, **c**) and phagocytic cups (**d**, **e**) using mEOS4b-GAS7b, with arrows indicating incorporated zymosan. Scale bar: 5 μm. **f–i** Super-resolution images showing the single-molecule localization of photoconvertible mEOS4b-tagged GAS7b. Left panels are the 60 nm slices of mEOS4b signals on glass (**f**), on liposomes (**g**), at the bottom of a phagocytic cup (**h**), at the top of a phagocytic cup (**i**) and at the indicated observation depths of the regions marked by red rectangles in **b**, **c** and **d**, **e**. Middle panels are enlarged images of the parts enclosed in red rectangles in the left panels. Graphs on the right show the occurrences of observed neighbour signals (shown in magenta) dependent on the neighbour distances within the regions enclosed by red rectangles in the left panels. The same numbers of signals from the simulated randomly placed GAS7b dimers and the simulated FFO sheet, under the assumption of the 10% observation of the total molecules, are shown in green and blue, respectively. The occurrence is described as the fold increase in the same numbers of the randomly distributed simulated signals. Dashed lines represent ±SD of 20 trial simulations. Scale bars: 1 μm (left) and 100 nm (middle)

relatively flat membranes, including lamellipodia, cell–cell junctions and membranes before their transformation into clathrin-coated pits. Lamellipodia contain the I-BAR domain protein IRSp53[35], as well as the F-BAR domain proteins CIP4[36], FBP17[37], srGAP1-4[38] and FCHSD1/2/Nwk[30]. Cell–cell junctions contain the I-BAR domain protein PinkBAR[24]. The relatively flat membrane before endocytosis contains FCHSD1/2[39] and FCHo1/2[40]. However, the means by which the BAR superfamily proteins form two-dimensional oligomers on these structures have remained enigmatic. Phagocytic cups are considered to be analogous structures to lamellipodia[1,41–43]. The studies of the BAR domain superfamily members involved in phagocytic cup formation have been limited to FBP17 in macrophages[44], IBARa in dictyostelium[45] and Bin2 in leucocytes[46]. None of these proteins has been suggested to form a two-dimensional oligomeric assembly in the phagocytic cup. Our data now indicate that the GAS7b sheet formation is critical in the membrane deformation during phagocytosis.

We showed that the N-surface is unlikely to be the membrane-binding surface of the GAS7 F-BAR domain upon the oligomer formation. Instead, GAS7 F-BAR likely binds to the membrane through the surface close to the side surface. The side surface was previously shown to be used by the F-BAR domain of FBP17 to bind to the flat membrane, presumably before it deformed the membrane into a tubular shape by binding through the concave N-surface[10]. GAS7 did not induce significant membrane deformation when it bound to flat membranes under our conditions. This might arise from the membrane binding preferentially on the side surface of the F-BAR domain, as determined by the mutagenesis analysis (Fig. 3, Supplementary Fig. 5a). Although GAS7 alone did not induce prominent membrane deformation of the already micron-sized GUVs, GAS7 F-BAR in HeLa cells was localized at micron-sized, cup-like assemblies (Supplementary Fig. 5a). Therefore, the GAS7 F-BAR domain itself might participate in micron-sized relatively flat membrane remodelling, under as yet unidentified conditions.

Most membrane structures involving BAR superfamily proteins are accompanied by Arp2/3 complex-mediated actin polymerization. The SH3 domains in the BAR superfamily proteins bind to the WASP family proteins, including N-WASP and WAVE, which then activate the Arp2/3 complex. GAS7b binds to N-WASP with the WW domain, and GAS7cb has the SH3 domains in addition to the WW domain[13,14]. These domains bind to N-WASP, thus regulating the Arp2/3 complex-mediated actin polymerization[47–49]. The F-BAR domain alone failed to rescue phagocytosis, suggesting the requirement of the WW domain in GAS7b to support phagocytosis. The membrane binding of GAS7 was also essential, as the proteins with mutations in the F-BAR domain, which inhibited membrane binding, also failed to restore phagocytosis. Therefore, the WASP family protein-mediated actin dynamics regulation may cooperate with GAS7 for phagocytic cup formation. The formation of phagocytic cups is considered to share mechanisms with lamellipodia,

Arp2/3-dependent protrusive structures for cell migration, in which the Arp2/3 complex regulators including N-WASP play a significant role. The Arp2/3-complex-mediated actin polymerization is also necessary for phagocytic cup formation[50]. Interestingly, GAS7-knockout cells exhibited defects in lamellipodia formation, indicating that GAS7 is part of the shared mechanisms between phagocytosis and lamellipodia formation, including actin dynamics regulation[41,43].

The BAR domain superfamily proteins assemble on the membrane and activate N-WASP-mediated actin polymerization[47–49]. The high-density assembly of the BAR superfamily proteins on the membrane results in the accumulation of other domains, including the WW and SH3 domains, to near milli-molar concentrations. The electron microscopic and super-resolution analyses support such a high concentration of GAS7 on the membrane. The actin polymerization by the Arp2/3 complex is enhanced by the highly concentrated N-WASP-binding domains on the membranes[49,51]. Further analyses will clarify how GAS7 sheet formation and actin polymerization cooperatively regulate membrane deformation during phagocytosis.

## Methods

**Gene cloning, protein purification and crystallization**. Mouse GAS7cb (GenBank accession XM_006532202.1) was cloned from a mouse brain complementary DNA library by PCR. The primers used in this study were listed in Supplementary Table 2. The DNA sequences encoding the F-BAR domains of mouse GAS7cb (166–476 aa) and GAS7cb (1–476 aa) were cloned into the pCold II vector (Takara Bio, Japan), using the BamHI and SalI restriction sites to include an N-terminal affinity tag (His₆). To express the selenomethionine (SeMet)-substituted protein, Escherichia coli (E. coli) strain JM109 (Takara Bio) was transformed with the expression plasmid for the F-BAR domain. Escherichia coli cells were grown in M9 minimal medium at 37 °C. When the OD₆₀₀ reached 0.7, a final concentration of 60 mg l⁻¹ L-SeMet; 100 mg l⁻¹ each of L-threonine, L-lysine hydrochloride and L-phenylalanine; and 50 mg l⁻¹ each of L-leucine, L-isoleucine and L-valine were added to prevent methionine production. The protein was overexpressed at 15 °C in the presence of 0.5 mM isopropyl-β-D-1-thiogalactopyranoside. The harvested E. coli cells were resuspended in buffer containing 20 mM Tris-HCl (pH 7.5), 0.50 M NaCl, 0.50 M MgCl₂, 10 mM 2-mercaptoethanol and 1 mM phenylmethanesulfonyl fluoride (PMSF), and were disrupted with an ultrasonic homogenizer. The lysate was centrifuged, and the supernatant was loaded on a Ni-Sepharose 6 Fast Flow column (GE Healthcare). The column was washed with 20 mM Tris-HCl (pH 7.5) buffer, containing 0.80 M NaCl, 10 mM 2-mercaptoethanol and 20 mM imidazole. The protein was eluted with 20 mM Tris-HCl (pH 8.5) buffer, containing 0.80 M NaCl and 0.30 M imidazole. The eluted protein was further purified on a HiLoad Superdex 200 pg 16/600 column (GE Healthcare), using 20 mM Tris-HCl (pH 7.5) buffer containing 0.20 M NaCl and 10 mM 2-mercaptoethanol, and was concentrated with an Amicon Ultra filter (Merck Millipore). The crystals of the F-BAR domain (13.3 mg ml⁻¹) were grown using the vapour-diffusion method, by 1:1 mixing of the protein solution with a reservoir composed of 12% PEG 4000, 4% PEG 8000, 0.16 M ammonium sulphate, 80 mM HEPES-NaOH (pH 7.5) and 20 mM MES (pH 6.5).

The E. coli strain Rosetta 2 was transformed with the expression plasmid for GAS7cb, and the protein was overexpressed in LB medium at 15 °C. The bacterial pellet was resuspended in buffer containing 20 mM Tris-HCl (pH 7.5), 0.80 M NaCl, 10% glycerol, 1 mM DTT and 1 mM pefablock or PMSF, and the bacteria were disrupted with an ultrasonic homogenizer. The lysate was centrifuged, and the supernatant was mixed with Ni-Sepharose and washed once in the batch mode.

The beads were then transferred to a column and washed with 150 ml of 20 mM Tris-HCl, pH 7.5, containing 0.80 M NaCl, 0.2 mM pefablock or PMSF, 1 mM dithiothreitol (DTT) and 20 mM imidazole. The protein was eluted with step gradients of 40, 60, 80 and 500 mM imidazole in 20 mM Tris-HCl (pH 7.5) buffer, containing 0.8 M NaCl, 0.2 mM pefablock or PMSF and 1 mM DTT. The GAS7cb fractions were further purified on a HiLoad Superdex 200 pg 16/600 column (GE Healthcare), using 20 mM Tris-HCl (pH 7.5) buffer containing 0.3 M NaCl and 1 mM DTT. The protein (15.0 mg ml$^{-1}$) was concentrated with an Amicon Ultra filter and then crystallized using the vapour-diffusion method, by 1:1 mixing with the crystal reservoir composed of 0.1 M sodium citrate, pH 5.0, 1.2 M sodium formate, 0.2 M NaCl and 1–4% ethylene glycol.

To express GAS7b, *E. coli* strain JM109 or BL21 was transformed with the pGEX6P1 plasmid, containing the mouse GAS7b (aa 62–476; GAS7cb amino-acid residue numbers) gene cloned into the *Bam*HI site. After immobilization of the protein on the Glutathione Sepharose (GE Healthcare) and washing, the GST tag was cleaved by Precision protease, and the cleaved proteins were collected in 10 mM Tris-HCl (pH 7.5) buffer, containing 150 mM NaCl and 1 mM EDTA[52].

To express the F-BAR domain of GAS7 for light and electron microscopy, *E. coli* strain JM109 or BL21 was transformed with the pGEX6P1 plasmid, containing mouse GAS7cb (166–476 aa) cloned into the *Bam*HI site. The purification was performed as for GAS7b. For electron microscopy, GAS7b, the F-BAR domain of GAS7 and GAS7cb were purified by gel filtration, in a similar manner to the proteins used for crystallization.

**Data collection and structure determination.** Crystals of the F-BAR domain or GAS7cb were soaked in a cryo-protective solution, composed of 22% PEG 4000, 70 mM ammonium sulphate, 80 mM HEPES-NaOH (pH 7.5) and 15.5% ethylene glycol or 25% ethylene glycol, respectively, and flash cooled in a nitrogen cryo-stream. Data sets were collected at the Photon Factory beamline BL-1A (Tsukuba, Japan) and the SPring-8 beamlines BL32XU, BL38B1 and BL44XU (Hyogo, Japan), and were processed with XDS[53] or HKL2000[54].

The structure of the GAS7 F-BAR domain was solved by the single-wavelength anomalous dispersion method, using the program autoSHARP[55]. The structure of GAS7cb was solved by the molecular replacement method, using the structure of the GAS7 F-BAR domain as the search model, with PHENIX[56]. Refinement was performed with PHENIX. Figures were created with the program PyMol (http://www.pymol.org). The Ramachandran plot analysis indicated that 95.6% (GAS7 F-BAR domain) and 78% (GAS7cb) of the residues were in the favoured regions and 1.5% (GAS7 F-BAR domain) and 3.5% (GAS7cb) of the residues were in the outlier regions.

**MD simulations.** A 1-μs MD simulation of F-BAR in solution (150 mM NaCl) was performed using Gromacs 2018.1[57,58]. The system was brought to thermodynamic equilibrium at 300 K and 1 atm, using the Nosé–Hoover thermostat and the Parrinello–Rahman barostat. The equations of motion were integrated with a time step of 2 fs. The long-range Coulomb energy was evaluated using the particle mesh Ewald method. The CHARMM36m force field[59] was used.

**Fluorescence microscopy of liposomes.** GUVs were prepared by natural swelling. The lipids, phosphatidylcholine (PC) (P3841, Sigma-Aldrich), phosphatidyletha-nolamine (PE) (P7693, Sigma-Aldrich), PS (P5660, Sigma-Aldrich) and PIP$_3$ (P-3916, Echelon Biosciences), were prepared in chloroform in a glass tube at the indicated molar ratios, to a final total concentration of 0.1 mM lipids, dried under nitrogen gas and subsequently incubated under vacuum. Subsequently, 20 μl of buffer containing 10 mM Tris-HCl (pH 7.5), 300 mM sucrose and 1.0 mM EDTA was added to the glass tube, which was then sealed and incubated at 45 °C for 8 min (prehydration). A 230 μl aliquot of the same buffer was added, and the tube was resealed and incubated at 37 °C for 2 h to produce the GUV suspension. After this incubation, the GUV suspension was mixed with buffer containing 10 mM Tris-HCl (pH 7.5), 1 mM EDTA and 150 mM NaCl in a 1:1 ratio, and then GFP-GAS7b and GUVs were incubated for 5 min at room temperature. The mixture was combined with 1% bovine serum albumin (BSA) as a blocking reagent, and observed with an FV1000D (Olympus) confocal microscope.

**Electron microscopy.** Lipid monolayers containing PC, PE and PS at a molar ratio of 1:1:3 were formed on the carbon-coated EM grid in buffer containing 50 mM Tris-HCl (pH 8.0) and 100 mM NaCl, and then covered with proteins[21,22]. The EM grids were then stained with uranyl acetate.

**Liposome co-sedimentation assays.** The in vitro liposome-binding analysis[23,60] was performed using the GAS7 in Fig. 2f, j, l, m or the GFP-GAS7 in Fig. 2g–i, k. Briefly, liposomes were made from the Folch fraction, a brain total lipid fraction rich in PS (Sigma-Aldrich, B1502). Liposomes were also made by mixing PC (Sigma-Aldrich, P3841), PE (Sigma-Aldrich, P7693), PS (Sigma-Aldrich P5660) and PIP$_3$ (P-3916, Echelon Biosciences). Lipids in chloroform were dried under nitrogen gas and subsequently incubated under vacuum. The dried lipids were resuspended in buffer containing 10 mM Tris-HCl (pH 7.5), 1 mM EDTA and 200 mM NaCl for Fig. 2g, k, in high-salt buffer containing 10 mM Tris-HCl (pH 7.5), 1 mM EDTA and 300 mM NaCl for Fig. 2h, and buffer containing 10 mM

Tris-HCl (pH 7.5), 1 mM EDTA and 150 mM NaCl for Fig. 2f, i, j, l, and then incubated for 1 h at 37 °C to form liposomes. Subsequently, 0.5 μM protein and 0.2 mg ml$^{-1}$ (Fig. 2f, l) or 0.4 mg ml$^{-1}$ liposomes (Fig. 2g–k) were incubated in the same respective buffers for 20 min at room temperature. The liposomes were then precipitated by centrifugation at 50,000 r.p.m. for 20 min in a TLA100 rotor (Beckman Coulter). The pellet and supernatant were fractionated by sodium dodecyl sulfate-polyacrylamide gel electrophoresis (SDS-PAGE) and visualized by staining with Coomassie Brilliant Blue.

**Crosslinking assays.** The BS(PEG)5 (PEGylated bis(sulfosuccinimidyl)suberate) linker, with spacer arm length of 21.4 Å, was used at the indicated concentrations. The proteins (0.5 μM), in buffer containing 10 mM HEPES (pH 8.0), 150 mM NaCl and 1 mM EDTA, were incubated at room temperature for 20 min at 37 °C with or without liposomes. Then, the proteins were mixed with BS(PEG)5 and were incubated for 10 min in 37 °C. The reaction was quenched by adding Tris-HCl (pH 7.5) to a final concentration of 25 mM, incubated for 15 min at room temperature and then analysed by SDS-PAGE and western blotting.

**Cell culture.** GAS7, ΔFFL1 (aa 171–197 deletion, GAS7cb residue numbers), ΔFFL2 (aa 209–216 deletion) and all of the amino-acid substitution mutants (all described in terms of the GAS7cb amino-acid residue number) were cloned into the pEGFP-C1 vector, and EGFP was substituted with its brighter variant, Venus[61]. The DsRed membrane was expressed using the pDsRed-monomer-Mem vector (Clontech). HeLa cells were cultured in Dulbecco's modified Eagle's medium (DMEM) supplemented with 10% fetal bovine serum (FBS)[52]. Transfection was performed with Lipofectamine LTX and PLUS reagents (Invitrogen), according to the manufacturer's instructions. Cells were observed with a confocal microscope (Olympus FV1000D).

RAW264.7 cells were cultured in DMEM supplemented with 10% FBS. Transfection of RAW264.7 cells ($1 \times 10^5$ cells per 10 μl) with siRNA (1 μl, 20 μM) was performed via electroporation with 1 pulse from a NEON transfection system (Invitrogen), at 1680 V and 20 ms. The GAS7 siRNA solution was a mixture of three siRNAs from Invitrogen (GAS7HSS144787, GAS7HSS144788 and GAS7HSS144789). The control siRNA was also from Invitrogen.

**Knockout and retrovirus-mediated gene transfer.** The CRISPR/Cas9 system was used as described previously[62]. The guide RNA targeting the first exon of GAS7b (GGCGGAGGGGGGACCATTCC) was designed using the server http://crispr.mit.edu[63] and inserted into the pX330 vector[62]. After transfection, the cells were cloned by monitoring the GFP fluorescence from the reporter plasmid pCAG-EGxxFP with the GAS7 genome fragment, using a fluorescence-activated cell sorter (FACSAria (BD)). GFP, GFP-GAS7, mEOS4b-GAS7b and GAS7 mutants were each introduced into the pMXs vector and expressed in GAS7-knockout cells, using a retrovirus produced by PLAT-A packaging cells[64]. The cells were cloned and isolated using a cell sorter. Clones with GAS7 expression similar to that of the parental cells were selected, and examined for their phagocytic activity.

**Phagocytosis assays.** Red-coloured zymosan (Invitrogen) or zymosan (Sigma) was opsonized with bovine serum albumin and incubated at 0.2 mg ml$^{-1}$ with cells at 37 °C. The cells were either subjected to live-cell imaging or fixed after 1 h. Live images were obtained with a confocal microscope (Olympus FV1000D) at 37 °C in a 5% CO$_2$ atmosphere.

Wild-type or GAS7-knockout RAW264.7 cells grown on cover glasses were incubated with red-coloured zymosan (Invitrogen) or red fluorescent latex beads (2 μm, Sigma) coated with purified rabbit IgG (Sigma), for 1 h at 37 °C. The cells were washed, fixed with 4% paraformaldehyde for 20 min and mounted. Approximately 100–200 cells from six to seven randomly selected fields were examined, to determine the number of incorporated particles.

**Immunofluorescent staining.** Cells grown on cover glasses were treated with BSA-coated zymosan beads for 30 min, and then fixed with 4% paraformaldehyde in the appropriate medium at room temperature for 20 min. The cells were permeabilized with 0.5% Triton X-100 in Tris-buffered saline (TBS) at room temperature for 20 min. The samples were blocked with 3% BSA and 10% goat serum in TBS containing 0.1% Triton X-100 (TBS-T) for 1 h, and then incubated with primary antibodies in Can Get Signal immunostain solution (TOYOBO) for 2 h. After washing with TBS-T, the cells were incubated with fluorescently labelled secondary antibodies and phalloidin for 1 h. After washing, the cells were mounted using Prolong Gold (Thermo Fisher/Invitrogen).

The following antibodies were used: mouse anti-GAS7 (clone 2F6, Origene, TA501756, 1:100 dilution); mouse anti-integrin-αM (clone Ox42, Santa Cruz Biotechnology, sc-53086, 1:100) and anti-CD206 (Santa Cruz Biotechnology, sc-376108, 1:100); rabbit anti-N-WASP (clone 30D10, Cell Signaling, 4848, 1:50); mouse anti-Arp3 (clone FMS338, Abcam, ab49671, 1:250); rabbit anti-GFP (MBL, 598, 1:1000); rat anti-GFP (clone GF090R, Nacalai, 040080, 1:1000); Alexa Fluor-conjugated highly cross-adsorbed anti-mouse or anti-rabbit antibodies (Thermo Fisher, A-11034, A-11036, A-21245, A-11029, A-11031, A-21236, 1:400) and Alexa Fluor-conjugated phalloidin (Thermo Fisher/Invitrogen, A12380, 1:100); and CF488A-labelled anti-rat IgG antibody (Biotium, 20023, 1:400).

Images of cells were obtained using an FV1000 laser-scanning confocal microscope (Olympus) equipped with a ×100 NA 1.45 oil lens (Olympus) at room temperature.

**Western blotting**. After SDS-PAGE, the proteins in the gel were transferred onto the membrane (Immobilon P, IPVH00010, Merck Millipore) by using Trans-Blot SD Semi-Dry Transfer Cell (Bio-Rad). The membrane was blocked with 5% skim milk in PBS supplemented with 0.05% Tween-20 (PBS-T). Then, the proteins were examined by mouse anti-GAS7 (Origene, 2F6, TA501756), anti-actin (Merck Millipore, MAB1501), anti-GFP (MBL, 598) and anti-glyceraldehyde-3-phosphate dehydrogenase (Santa Cruz Biotechnology, sc-166574) antibodies as primary antibody at 1:1000 dilution, followed by the secondary antibody of anti-mouse or rabbit IgG alkaline phosphatase conjugate (Promega) in PBS-T with 1:10000 dilution. The proteins were detected by 5-bromo-4-chloro-3-indoryl phosphate/nitroblue tetrazolium (Roche).

**FRAP and time-lapse imaging**. Cells on coverslips were placed in the microscope's (FV1000, Olympus) humidified chamber with 5% $CO_2$. The region of interest was bleached by a 473 nm laser for 1 s. The fluorescence intensities were analysed by the Image J (NIH) software. The fluorescence intensity before bleaching was set to 100%. Fluorescence after bleaching was considered to be 0%.

For the analysis of lamellipodia, cells were transfected with pCAH-Lifeact-EGFP[65] using Lipofectamine 3000 (Invitrogen), cultured for 1 day and then cultured in glass-bottom dishes (IWAKI) for 1 day. Time-lapse imaging was performed using an FV1000 (Olympus) laser-scanning confocal microscope at 30 s intervals.

**Super-resolution microscopy observation**. The photoconvertible fluorophore, mEOS4b, was used because it can be converted to a red fluorescent protein by ultraviolet (UV) irradiation, even after fixation[33]. For the observation of mEOS4b-GAS7b on liposomes, 50 μl portions of liposomes, composed of 0.125 mg ml⁻¹ lipids with 1% of biotin-PE (Avanti) in 300 mM sucrose, were placed on glass-bottom dishes coated with streptavidin and biotin-BSA for immobilization, as described previously[37]. The liposomes were overlaid with 10 μl of 4 μM mEOS4b-GAS7b in 10 mM Tris buffer (pH 7.5), containing 150 mM NaCl and 1 mM EDTA, and sandwiched with a cover glass. After 15 min at room temperature, the liposomes were fixed with 4% paraformaldehyde and 0.2% glutaraldehyde in HEPES-buffered saline for 20 min.

GAS7-knockout RAW264.7 cells expressing mEOS4b-GAS7b were cultured on glass coverslips and incubated with zymosan for 1 h. The cells were then fixed in 4% paraformaldehyde with 0.2% glutaraldehyde (electron microscopy grade) in HEPES-buffered saline, containing 30 mM HEPES (pH 7.5), 100 mM NaCl and 2 mM $CaCl_2$ for 20 min.

The fixed samples were reduced with PBS (0.1% $NaBH_4$) for 5 min. The samples were then stored in PBS with 1% polyvinyl alcohol and 10 mM cysteamine (MEA). The samples were sealed with a 1:1:1 vaseline lanolin paraffin mixture or vaseline and then imaged using the N-STORM setup with an iXon DU-897E electron-multiplying charge-coupled device camera (Andor). The images were recorded with the NIS-elements software version 4.60.00 and N-STORM version 4.0.0.215, using the 3D-STORM mode. Images ($10 \times 10^5$) were acquired continuously with a 12 mW 543 nm laser, with continuous activation by a 405 nm laser. mEOS4b emits stochastic signals as it is converted from green to red by UV or violet light activation[66–68]. Red signals were erased upon observation by the 543 nm laser.

**Super-resolution microscopy data analysis**. Each single-molecule localization was analysed using the NIS-elements software provided by Nikon. Signals in the same pixel (160 nm square) in continuous images were considered to be derived from the same molecule and then merged. The $xy$ drifts between $z$ stacks were collected, using either fiducial markers or autocorrelation between images. The coordinates of the signals were exported and drift corrected using ThunderSTORM[69].

The visualization of molecules was performed using the MATLAB software (Mathworks). Molecule localization was shown by the probabilities of the mixed Gaussian distribution of the signals, where the accuracy of each signal was used as the deviation (sigma) for the mixed Gaussian distribution. The deviation in the depth direction ($z$) was set to be twice that in the focal plane ($xy$).

The randomness of the signals was analysed using MATLAB with a modified Ripley's $k$ function analysis[70]. The region of interest was selected, and the distances of the observed signals to their neighbours were calculated using the three-dimensional coordinates of the signals. The frequency of the neighbours in 5 nm step distances was calculated (i.e., the numbers of neighbours from $5 \times i$ to $5(i + 1)$ nm distances from a signal were counted, $i = 2–9$). The same number of randomly placed signals was generated as simulations, which were used for determining the fold increase in the observed neighbours as compared to the neighbours of random signals. The random signals were simulated 20 times, and these frequencies of neighbours were used to estimate the fold increases in the number of observed neighbours, as shown with ±SD. Frequencies below 10 nm distances were not shown because of the small number of signals, which result in a large variance in the fold increase.

The simulation for the comparison to the observed super-resolution signals was performed using the same number of signals as that of the observed signals to be compared. The simulated dimers were generated by pairs of labels with a distance of 11 nm. The simulated FFOs were the string of dimers with a spacing of 5 nm and a 40° tilt, as in the FFOs in the crystal. To generate the simulated FFO sheets, the FFOs were aligned in parallel, but the anti-parallel alignments gave almost identical results in the comparison. In total, 40% of the molecules were considered to form sheets, whereas the rest of the molecules were random dimers. We estimated the averaged concentration of the proteins and the number of proteins in the cells by western blotting using the purified GAS7b as the standard, and these values were compared with the total number of signals in RAW264.7 cells. Consequently, 10% of the labels were assumed to be observed. Each signal had an accuracy of 20 nm in the simulation.

**Statistical analyses**. All data are expressed as the mean ± SD, as indicated in the legends. Data for each condition were obtained from at least three independent experiments. Statistical analyses were performed using Microsoft Excel and Student's $t$ test. A value of $P < 0.05$ was considered significant.

**Reporting summary**. Further information on research design is available in the Nature Research Reporting Summary linked to this article.

## Data availability
Atomic coordinates and structure factors for the crystal structures of the GAS7 F-BAR domain and GAS7cb have been deposited in the Protein Data Bank, under the accession codes 6IKN and 6IKO, respectively. The source data underlying Figs. 2f–m, 3i–l and Supplementary Figs. 1b–c, 4a–h, 6, 7d–f, h and i are provided as a Source Data File. Other data are available from the corresponding authors on reasonable request.

## Code availability
The MATLAB codes and the source coordinates for Fig. 4 are available at GitHub .

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

## Acknowledgements

We thank Prof. Oliver Daumke (Max-Delbrück-Centrum for Molecular Medicine) for critical reading of the manuscript. We thank Prof. Toshio Kitamura (The University of Tokyo) for the PLAT-A cells, Prof. Masahito Ikawa (Osaka University) and Prof. Taro Kawai (Nara Institute of Science and Technology) for the CRISPR/Cas9 system, Dr. Yohei Katoh and Prof. Kazuhisa Nakayama (Kyoto University) for the nanobody to GFP, Dr. Takahiro Fujiwara, Dr. Rinshi S. Kasai (Kyoto University) and Dr. Kenichi G.N. Suzuki (Gifu University) for advice on STORM and PALM, Dr. Kouta Mayanagi (Kyushu University), Dr. Hideki Shigematsu (RIKEN SPring-8 Center) and Ryo Ugawa (Laboratory for Technical Support, Medical Institute of Bioregulation, Kyushu University) for their support and advice on electron microscopy and Dr. Arthur Melo,

Ayumi Takemoto, Kuniko Ohtake, Kosaku Kamihara, Ayana Wada, Katsuya Inamine, Yuki Kawasaki, Seiichiro Hayashi and all of the members of the laboratories for technical assistance and helpful discussions, and the staff members of the beamlines, Dr. Kunio Hirata (RIKEN SPring-8 Center), Dr. Naoki Sakai (RIKEN SPring-8 Center), Dr. Seiki Baba (Japan Synchrotron Radiation Research Institute) and Dr. Takashi Kumasaka (Japan Synchrotron Radiation Research Institute) and others, for assistance with our data collection. Synchrotron radiation experiments were performed at the Photon Factory with the approval of the Photon Factory Program Advisory Committee (Proposal Nos. 2011G092 and 2012G691) and at SPring-8 beamlines BL32XU, BL38B1 and BL44XU. This work was supported by grants from the Funding Program for Next Generation World-Leading Researchers (NEXT program LS031), JSPS (KAKENHI 26291037, JP15H0164, JP15H05902, JP17H03674, JP17H06006), JST CREST (JPMJCR1863), Osaka Cancer Research Foundation, the Naito Foundation, the Sumitomo Foundation, the Mitsubishi Foundation, the Sagawa Foundation for Promotion of Cancer Research and the NAIST Interdisciplinary Frontier Research Project to S.S., JSPS (KAKENHI 24687014, 25121726, JP17K07309) to A.S., JSPS (KAKENHI JP16K07351) to K.H.-S., JSPS (KAKENHI 23121507) to Y.I., JSPS (KAKENHI JP15H04357, JP17KT0026), MEXT 'Priority Issue on Post-K Computer' (Building Innovative Drug Discovery Infrastructure through Functional Control of Biomolecular Systems) to A.K. and Platform Project for Supporting Drug Discovery and Life Science Research (Basis for Supporting Innovative Drug Discovery and Life Science Research (BINDS)) from Japan Agency for Medical Research and Development (AMED), under Grant Numbers JP19am0101070 and JP19am0101072. The computations were partly performed using the supercomputers at the RCCS, The National Institute of Natural Science, ISSP and The University of Tokyo. This research also used the computational resources of the K computer, provided by the RIKEN Advanced Institute for Computational Science through the HPCI System Research project (Project ID: hp180201). This work was partly performed in the Cooperative Research Project Program of the Medical Institute of Bioregulation, Kyushu University.

## Author contributions

K.H.-S., Y.I., K.Takeshita and A.S. collected the X-ray diffraction data and determined the crystal structures. K.Takemura and A.K. performed MD simulations. K.H.-S., S.K., M.A.F., W.N.I.W.M.N., T.H.N.T., T.I. and K.Takeshita, expressed the proteins and performed biochemical analyses. N.M., M.A.F., K.I., S.S. and A.S. performed the electron micrographic observations. T.N., S.H.-N., K.O.-Y. and S.S. performed the cell biological analyses. K.H.-S., K.O.-Y., M.T. and S.S. prepared and analysed the super-resolution imaging samples. S.S., A.K., D.K., M.Y. and A.S. designed the research and supervised the project. K.H.-S., Y.I., M.A.F., T.N., K.Takemura, A.K., A.S. and S.S. wrote the manuscript with input from all other authors.

## Competing interests

The authors declare no competing interests.
