## [Peer Review File · Nature Communications]

Reviewers' comments:

Reviewer #3 (Remarks to the Author):

The authors have addressed most of my concerns in the revised version. The super-resolution data is more clearly presented and the conclusions drawn seem justified.

Reviewer #5 (Remarks to the Author):

This study resolve the molecular structure of GAS7 and the oligomerization details of GAS7. The structure data is strong and novel, reporting the new feature of F-BAR domain protein in sheet formation. Furthermore, the work provide evidence that GAS7 is involved in phagocytosis. However, the structural link and GAS7 oligomeration with the cellular function are not strongly supported. The results indicated that the interplay between GAS7 and actin polymerization is important for phagocytosis but it is not clear how protein sheets formed by GAS7 oligomerization play a key role in phagocytosis. The authors have performed extensive experiments according to reviewers' comments. However, the main points are still missing.

1.The loop mutants have diminished membrane binding of GAS7. Moreover, the R326A mutant displays less membrane binding affinity. Based on these experiments, the authors concluded that the oligomerized GAS7 has higher affinity to membranes than non-oligomeric GAS7. However, it is possible that the loops and R326 are involved both in membrane association and oligermization of GAS7. There are several positively charged residues in the loops, particularly in loop 2 (KK), which could be the membrane binding sites. It is known that the BAR domain proteins possess multiple membrane binding sites. Thus, this conclusion is not properly addressed. Similar conclusion was made in the phagocytosis experiments. Due to lack of this important evidence, the role of GAS7 oligomerization in phagocytosis is rather preliminary.

2.The full length GAS7 rescued the phagocytosis defects caused by GAS7 knockout. However, F-BAR alone did not rescue the phagocytosis defects significantly suggesting that interaction of N-WASP with WW domain of GAS7, which is not within the BAR domain region is important for phagocytosis, and the membrane interaction of GAS7 is not essential for phagocytosis. This is further supported by the experiments with overexpressed Lifeact in GAS7 knockout cells, showing that the frequency of lamellipodia was decreased in knockout cells.

3.The overexpressed GAS7 oligomerized by themselves into patches in HeLa cells. These patches could be overexpression artifacts because of missing interaction partners in HeLa cells, which does not express endogenous GAS7. On the other hand, the phagocytosis data in macrophages are more convincing.

4.GAS7 didn't induce membrane deformation in giant vesicles and but flexible striations were found on the membranes. However, it is important to make sure that the observed GUVs are unilamellar vesicles. These type of membrane striations are quite common in multilamellar vesicles because the giant vesicle system is very heterogeneous, a mixture of unilamellar and multilamellar vesicles. Thus, it is important to know that these striations are formed by GAS7, not membrane layers. The fluorescent GAS7 can be used in this case to exclude this possibility. Quantification of striation formation in control and GAS7 bound liposomes would also be helpful to support the conclusion. Furthermore, why GAS7 did not cause striations in smaller vesicles?

5.The study indicated that FFL2 is not likely to be involved in membrane insertion. However, membrane insertion of GAS7 and FFL2 mutants should be examined because high salt did not abolish the membrane interaction completely, suggesting that hydrophobic interaction is involved.

Reviewer #6 (Remarks to the Author):

This report by Hanawa-Suetsugu is the result of an integrated study to understand the mechanistic basis of GAS7 contributions to phagocytosis. The authors combine data from x-ray crystallography,

cryoEM, in vitro and in vivo fluorescence, and a number of other approaches to support their hypothesis that lateral assembly of GAS7b sheets on flat membranes is a key intermediate for driving formation of the phagocytotic cup in macrophages. The attempted degree of methodological integration is impressive, and the subject of the study is extremely relevant as progress in mechanistic understanding related to membrane remodeling is quite slow due to the complexity of the processes involved.

In response to the previous reviews, the authors have made a number of additions that partially address concerns raised by the referees. However, some of the key concerns were not properly addressed, leaving a lot of open ends and disconnects within the study. Overall, the data support that some form of lateral assembly of GAS7 contributes to phagocytosis, but anything beyond that seems reaching. This is saddening because clearly, the authors have put a lot of effort into this study. That said, it also is sad that in planning and executing the studies, little attention was paid to details that would help to make data internally consistent and interpretable. This oversight leads to a collective of data that in their entirety fail to (fully) support the claims that are made throughout.

Detailed comments - there are major issues with this work:

Crystallographic studies: these studies are, of course, interesting with interesting incremental new details about variations in the BAR-theme but since the SH3 and WW domains were not resolved, the big prize is still out there to be captured. Were these domains disordered to the point where there was absolutely no density at all? If not, it'd be useful to see how the "mass" of these domains spatially relates to the lattice model that is shown in Fig 1b (and extended Fig 2f). If the domains were too disordered to show at all, then how do the authors know that the crystals were formed from the full-length protein? Have the authors run a gel from their crystals? Apart from that: how is the double-layered model relevant later on? Bringing this up without relating it back to the in vivo data does not make much sense. In fact – in Fig 4 a different "sheet model" is used to put data into perspective. This switch is internally inconsistent and does not help.

CryoEM: the data provided in Fig 1e-j are pointless, and do not support any of the mechanistic claims or in vivo results. If anything, these images contradict the in vivo data. Fig 1f-h focus on weird membrane folds that happen to have dimensions roughly similar to the X-ray model. Fig 1i,j show some randomly crashed out protein aggregates, and what exactly Fig 1f is supposed to show/support is not clear. This looks like multilamellar structures, not "striations", and such structures can be formed by myriads of conditions, none of which have anything to do with what this study is about. In the previous review, referee #2 rightly pointed out that these images are not helpful, and that flat 2D-arrays of F-BAR domains have been reported in more clarity in the past. The authors did not address this issue, but rather try to make a hand-waving argument. To be clear: that reviewer did not ask for a detailed 2D-crystallographic analysis or tomographic data (as was done in previous studies by others) – the point raised in this previous review simply was that micron-sized arrays of any kind (even if they are highly disordered) are something one could not possibly miss in either negative stain or cryoEM. Trying to get out of this by invoking higher dynamic turnover doesn't seem the best idea because dynamic effects are reduced to exactly zero once the sample is either stained or frozen. Point is: Fig 3 suggests that these patches are indeed micron sized and given current detectors used in (cryo)EM, these things would indeed be impossible to miss if assembled on GUVs (yet nothing sensible shows in the figures that are provided). The statement that FRAP studies show high dynamic turnover is also somewhat troubled because extended Fig 8 top row suggests that recovery on GUVs is quite slow (and apparently "0" in HeLa cells – bottom row). Another issue here is that the lateral packing interactions within the lattice model proposed in Fig 1 are quite extensive....this would suggest that these aggregates should show some sort of structure when assembled on the membrane surface. Unless the authors have supportive data, it would be better to not use cryoEM data at all because as is, they are quite confusing and detrimental. But then: cryoEM is, by far, the best direct approach to support the author's claims.

Lipid substrates and assay conditions used: this is, perhaps, one of the biggest design flaws in the study.

- GUVs were made from PS/PE/PC mixes (some of which have unknown acyl chain compositions) in the absence of any salt;

- liposome binding studies used a Folch-Fraction (which one?) from Avanti (no catalogue number is provided and checking their catalogue – Avanti doesn't offer a "Folch"-Fraction, just a brain lipid extract with 10%PS content...not too high) and buffers with rather high salt concentrations (200 and 300mM respectively);
- in vivo assays use, well natural and asymmetric bilayers.

This lack in consistency seems to be a fatal flaw because the remodeling behavior of BAR-domain proteins is extremely sensitive to lipid-compositions, acyl chain lengths of lipids, salt, type of monovalent cation, counterions, and temperature – among a few parameters that are critical. Granted: doing these studies is challenging and any choice of lipids is, to some extent, arbitrary. That said: choices should be consistent across experimental approaches, and – more importantly – should be put into a "physiological perspective" at some point. None of that happens here and consequently, the in vitro data cannot reliably be correlated with any of the in vivo imaging studies, not can the in vitro studies be directly compared to each other. These limitations reduce the work to a pure phenomenological study that cannot support detailed mechanistic models.

R326A mutant: at some point the authors want to make up their mind how to utilize the results obtained with this mutant. As is, this is an incredibly messy and inconsistent story that – if anything – says that lateral assembly has little to nothing to do with phagocytosis. This mutant

- apparently binds lipids (though Fig 1 f/g are not consistent with the narrative given in the text or with what is shown in extended figure 4),
- but doesn't aggregate (?) (extended figure 6, Fig 2j, Fig 3a, and the narrative are all over the place, not consistently backing up each other),
- yet apparently can rescue some phagocytosis that in the author's model depends on lateral sheet formation.

Fig 2: for these data to match the narrative would require these experiments to be performed in solution AND in the presence of a lipid substrate.

Fig 3: panels e-g are inconsistent with panel dwhy is GAS7 concentrated on only one side in panel d while equally distributed all around in the other cases?

Less major:

Fig 1d: what is the weird feature at the top of the GUV? Seems that lateral assembly starts at a curved region?

Figure 3h: GAS7cb mutant is not shown but referred to in the text.

At some point early on, the authors might want to explicitly state

Reviewer #5 (Remarks to the Author):

This study resolve the molecular structure of GAS7 and the oligomerization details of GAS7. The structure data is strong and novel, reporting the new feature of F-BAR domain protein in sheet formation. Furthermore, the work provide evidence that GAS7 is involved in phagocytosis. However, the structural link and GAS7 oligomerization with the cellular function are not strongly supported. The results indicated that the interplay between GAS7 and actin polymerization is important for phagocytosis but it is not clear how protein sheets formed by GAS7 oligomerization play a key role in phagocytosis. The authors have performed extensive experiments according to reviewers' comments. However, the main points are still missing.

We appreciate your kind consideration of our manuscript. According to your comments, we modified our manuscript by omitting some data, including the cryoEM results. We also added some additional experiments, including the novel D207R mutant, which retains the membrane binding ability without intact oligomer formation, and clearly showed the importance of oligomer formation by GAS7 for phagocytosis. Furthermore, we added the EM of the monolayered flat membrane with the striations of GAS7, to clearly indicate the formation of the GAS7 sheet.

1. The loop mutants have diminished membrane binding of GAS7. Moreover, the R326A mutant displays less membrane binding affinity. Based on these experiments, the authors concluded that the oligomerized GAS7 has higher affinity to membranes than non-oligomeric GAS7. However, it is possible that the loops and R326 are involved both in membrane association and oligomerization of GAS7. There are several positively charged residues in the loops, particularly in loop 2 (KK), which could be the membrane binding sites. It is known that the BAR domain proteins possess multiple membrane binding sites. Thus, this conclusion is not properly addressed. Similar conclusion was made in the phagocytosis experiments. Due to lack of this important evidence, the role of GAS7 oligomerization in phagocytosis is rather preliminary.

As this reviewer properly pointed out, R326 could be involved in direct binding to the membrane, by using its positive charge to bind to the negatively-charged membrane. Thus, to avoid confusion, we omitted the data on the R326A mutant and instead examined another mutant, D207R. An MD simulation suggested that D207 is involved in the stability of the FFL2 loop, because D207 contacted the other residues of the F-BAR domain (Fig. S3). The D207R mutation does not reduce the positive charge, and therefore, it is not considered to affect the oligomer-independent membrane binding. Actually, in the liposome co-sedimentation assay, the D207R mutant had similar membrane binding ability to GAS7b (Fig. 2l), while it exhibited reduced oligomer formation *in vitro*, by a chemical cross-linking analysis (Fig. 2m), and abolished the sheet formation in HeLa cells (Fig. 3a). The D207R mutant failed to restore the phagocytosis cup formation when it was expressed in GAS7-knockout macrophages (Fig. 3h,i). Therefore, in combination with the results of the delta FFL2 mutant, the stability of FFL2 is strongly suggested to be involved in oligomer formation, and the oligomer formation is strongly suggested to be involved in the phagocytic cup formation.

To study the loop functions, we made the K209E and K208A/K209A mutants, which both showed reduced but detectable membrane binding abilities (Fig. 2i, j). Interestingly, these mutants retained the sheet formation ability, although the sheet formation by these mutants was weaker than that of GAS7b when overexpressed in HeLa cells (Fig. 3a). The K208E mutant could not restore the phagocytosis (Fig. 3h, i), and therefore the membrane binding of the loop is thought to be essential for the GAS7-mediated phagocytic cup formation.

Regarding the possible insertion of FFL2 into the membrane, the introduction of an arginine to replace Glu212 (Q212R mutant) at the tip did not reduce the membrane interaction, but rather increased it, strongly suggesting that FFL2 is not inserted into the membrane. The introduction of hydrophilic amino-acid residues to the hydrophobic loop of PACSIN/Syndapin reportedly inhibited membrane binding (Shimada et al., 2010; Wang et al., 2009). Therefore, the FFL2 of the GAS7 F-BAR domain is not likely to be the membrane insertion region, but instead represents one of the membrane contact sites.

We also made several additional mutants to identify the membrane binding surface of the GAS7 F-BAR domain. As shown in Figs. 2a and S5, the amino-acid residues that are involved in membrane binding were mapped to the region between the bottom surface (N-surface) and the side surface of the F-BAR domain, suggesting the formation of a binding surface equivalent to the FFO surface, the oligomer surface observed in the crystals of the GAS7 F-BAR domain.

2. The full length GAS7 rescued the phagocytosis defects caused by GAS7 knockout. However, F-BAR alone did not rescue the phagocytosis defects significantly suggesting that interaction of N-WASP with WW domain of GAS7, which is not within the BAR domain region is important for phagocytosis, and the membrane interaction of GAS7 is not essential for phagocytosis. This is further supported by the experiments with overexpressed Lifeact in GAS7 knockout cells, showing that the frequency of lamellipodia was decreased in knockout cells.

Most of the BAR domain-containing proteins have the SH3 domain, and are considered to cooperate with the SH3 domain interacting proteins, including N-WASP and dynamin, linking the membrane curvatures to the actin cytoskeleton or dynamin-mediated membrane scission (Takenawa and Suetsugu, 2007). Therefore, it is quite reasonable that the F-BAR domain alone could not restore the lack of GAS7, as in the cases of other BAR domain proteins. We included a description of this point in the discussion.

Furthermore, we would like to point out that the essentiality of the binding of the GAS7 F-BAR domain to the membrane in phagocytosis has been shown with several membrane-binding defective mutants of the F-BAR domain (Δ FFL2, D207R, and K209E) (Fig. 3h-j).

3. The overexpressed GAS7 oligomerized by themselves into patches in HeLa cells. These patches could be overexpression artifacts because of missing interaction partners in HeLa cells, which does not express endogenous GAS7. On the other hand, the phagocytosis data in macrophages are more convincing.

We completely agree that the localization of GAS7 in HeLa cells may represent an overexpression artifact. However, the results clearly showed the assembly of GAS7 on the membrane, and the behavior of these GAS7 patches resembled the phagocytic cup formation, and therefore, the overexpression analysis in HeLa cells is considered to be a good model system. Furthermore, the mutants that did not form patches in HeLa cells did not restore the phagocytic cup formation in GAS7 knockout cells, and thus the localization in HeLa cells is considered to clarify our understanding of GAS7. Therefore, we would like to keep the HeLa cell data as described in the revised manuscript. As the limitations of the experiments in HeLa cells may not have been clear in the text, we included a sufficient description regarding this point in the text (lines 230-233) as follows:

The assembly and the membrane localization of GAS7 were examined in HeLa cells, which do not express any isoforms of endogenous GAS7 (Supplementary Fig. 1b) and therefore could be a good model system to examine the assembly of GAS7 within cells.

4. GAS7 didn't induce membrane deformation in giant vesicles and but flexible striations were found on the membranes. However, it is important to make sure that the observed GUVs are unilamellar vesicles. These type of membrane striations are quite common in multilamellar vesicles because the giant vesicle system is very heterogeneous, a mixture of unilamellar and multilamellar vesicles. Thus, it is important to know that these striations are formed by GAS7, not membrane layers. The fluorescent GAS7 can be used in this case to exclude this possibility. Quantification of striation formation in control and GAS7 bound liposomes would also be helpful to support the conclusion. Furthermore, why GAS7 did not cause striations in smaller vesicles?

We included Rhodamine-labeled PE within the GUVs, and showed that the GUVs are very likely to be unilamellar, as indicated by the Rhodamine signals (Fig. 1a). It was difficult to obtain sufficiently clear EM images using GUVs due to several experimental limitations, such as thickness by the largeness of the GUVs, which induces a large amount of ice incorporation. It was also possible that the striations of the MLVs were observed, instead of the GAS7 striations in the previous cryoEM analysis. Therefore, we

replaced the previous cryoEM data with the images of a negatively stained lipid monolayer on the flat support of EM grid in the presence of GAS7, showing the striations of GAS7, which suggested the sheet-like assembly of GAS7 (revised Fig. 1c, d). The results clearly showed that the striations occurred on the membrane (Fig. 1c). Therefore, we believe that the GAS7 on the membrane forms the sheet-like two-dimensional assembly.

5. The study indicated that FFL2 is not likely to be involved in membrane insertion. However, membrane insertion of GAS7 and FFL2 mutants should be examined because high salt did not abolish the membrane interaction completely, suggesting that hydrophobic interaction is involved.

The residues in FFL2 are mostly hydrophilic (Supplementary Fig. 1d). Thus, it seems unlikely that this loop penetrates into the membrane. Especially, Gln212 is located at the tip of FFL2, and the Q212R mutant exhibited the membrane binding, indicating that at least the tip of FFL2 was not inserted into the membrane. We used 300 mM NaCl as the high salt condition, and 200 mM NaCl in the control experiment. Thus, we did not use an extremely high salt concentration in the high salt experiment. Therefore, it seems reasonable that the membrane interaction was not completely abolished in the high salt conditions. The membrane binding of GAS7 was absent at 500 mM NaCl; however, we did not include these data because the protein might also be misfolded at 500 mM NaCl. In addition, we could not make a mutation that supports the strong involvement of the hydrophobic interaction, because the amino-acid sequence of FFL2 is ADKKDPQGNGTV. Therefore, we would like to conclude that the FFL2 is not likely to be the membrane insertion loop.

Refs:

Shimada, A., Takano, K., Shirouzu, M., Hanawa-Suetsugu, K., Terada, T., Toyooka, K., Umehara, T., Yamamoto, M., Yokoyama, S., and Suetsugu, S. (2010). Mapping of the basic amino-acid residues responsible for tubulation and cellular protrusion by the EFC/F-BAR domain of paccin2/Syndapin II. *FEBS Lett* 584, 1111-1118.

Wang, Q., Navarro, M.V., Peng, G., Molinelli, E., Lin Goh, S., Judson, B.L., Rajashankar, K.R., and Sondermann, H. (2009). Molecular mechanism of membrane constriction and tubulation mediated by the F-BAR protein Paccin/Syndapin. *PNAS USA* 106, 12700-12705.

Takenawa, T., and Suetsugu, S. (2007). The WASP-WAVE protein network: connecting the membrane to the cytoskeleton. *Nat Rev Mol Cell Biol* 8, 37-48.

Reviewer #6 (Remarks to the Author):

This report by Hanawa-Suetsugu is the result of an integrated study to understand the mechanistic basis of GAS7 contributions to phagocytosis. The authors combine data from x-ray crystallography, cryoEM, in vitro and in vivo fluorescence, and a number of other approaches to support their hypothesis that lateral assembly of GAS7b sheets on flat membranes is a key intermediate for driving formation of the phagocytotic cup in macrophages. The attempted degree of methodological integration is impressive, and the subject of the study is extremely relevant as progress in mechanistic understanding related to membrane remodeling is quite slow due to the complexity of the processes involved.

In response to the previous reviews, the authors have made a number of additions that partially address concerns raised by the referees. However, some of the key concerns were not properly addressed, leaving a lot of open ends and disconnects within the study. Overall, the data support that some form of lateral assembly of GAS7 contributes to phagocytosis, but anything beyond that seems reaching. This is saddening because clearly, the authors have put a lot of effort into this study. That said, it also is sad that in planning and executing the studies, little attention was paid to details that would help to make data internally consistent and interpretable. This oversight leads to a collective of data that in their entirety fail to (fully) support the claims that are made throughout.

We appreciate your careful consideration of our manuscript. Please find our point-by-point responses below.

Detailed comments - there are major issues with this work:

Crystallographic studies: these studies are, of course, interesting with interesting incremental new details about variations in the BAR-theme but since the SH3 and WW domains were not resolved, the big prize is still out there to be captured. Were these domains disordered to the point where there was absolutely no density at all? If not, it'd be useful to see how the "mass" of these domains spatially relates to the lattice model that is shown in Fig 1b (and extended Fig 2f). If the domains were too disordered to show at all, then how do the authors know that the crystals were formed from the full-length protein? Have the authors run a gel from their crystals? Apart from that: how is the double-layered model relevant later on? Bringing this up without relating it back to the in vivo data does not make much sense. In fact – in Fig 4 a different "sheet model" is used to put data into perspective. This switch is internally inconsistent and does not help.

As for the structure of full-length GAS7cb, we first tried to obtain better crystallographic data than those presented in the previous version, to clearly observe the electron densities for the SH3 and WW domains. We improved the resolution of the full-length GAS7cb structure to 3.2 Å. However, the electron densities for these domains were still not clearly visible, and thus the masses of these domains could not be identified in the electron density maps. Therefore, we would like to keep the previous crystallographic analysis statistics. Based on the SDS-PAGE gel pattern of the crystals, there might be some degradation of the full-length protein in the crystal (Supplementary Fig. 2c). Nevertheless, the majority of GAS7cb molecules in the crystal seem to be intact, as determined by the SDS-PAGE gel pattern (Supplementary Fig. 2c). In addition, the crystal of GAS7cb possessed an unusually high solvent content (> 85%), indicating that the full-length GAS7cb protein is crystallized without the visibility of the SH3 and WW domains. Therefore, we would like to include the crystallographic data of the F-BAR domain of GAS7 that were determined using the GAS7cb crystals, for showing the consistency of the F-BAR domain structure in the various splice isoforms.

We apologize for the confusion originating from our drawing in the previous Figure 1. The double layered model that was shown in Figure 1 in the previous version showed the actual alignment of the F-BAR domains in the crystals, where there are two identical F-BAR alignments in different orientations. These two were shown as the double-layered model in our previous manuscript (previous Figure 1). We realized this representation was quite misleading, because the two pink and blue F-BAR alignments were identical to each other. Therefore, we removed one of the representations of the F-BAR alignment in the revised manuscript (revised Figure 2).

The sheet model in Figure 4 is based on the striations observed by the EM analysis, which showed the striations with ~5 nm spacing in the revised manuscript (Fig. 1). The dense assembly of GAS7 can be achieved by the alignment of the possible GAS7 filaments. Therefore, we proposed the sheet mode

in Fig. 4a. The sheet of GAS7 appeared to occur by the superresolution signals (Please look at the enlarged panels in Figure 4g, i). We agree that the previous EM analysis had problems, in that it could be the possible observation of the multilamellar vesicles. Therefore, in this revised manuscript, we replaced it with the EM analysis of the mono-layered membrane decorated with GAS7b, which clearly shows the sheet-like striations of GAS7.

The WW and SH3 domains can be positioned on one side of the FFO, because there is a huge space between the F-BAR domains of FFO for the linker region between the F-BAR and the WW domains to pass through. To show the sufficient space for the WW and SH3 domains to gather on the membrane-free side of the FFO, we presented the hypothetical model of the WW domains placed on the FFO (revised Fig. 2c).

CryoEM: the data provided in Fig 1e-j are pointless, and do not support any of the mechanistic claims or in vivo results. If anything, these images contradict the in vivo data. Fig 1f-h focus on weird membrane folds that happen to have dimensions roughly similar to the X-ray model. Fig 1i,j show some randomly crashed out protein aggregates, and what exactly Fig 1f is supposed to show/support is not clear. This looks like multilamellar structures, not "striations", and such structures can be formed by myriads of conditions, none of which have anything to do with what this study is about. In the previous review, referee #2 rightly pointed out that these images are not helpful, and that flat 2D-arrays of F-BAR domains have been reported in more clarity in the past. The authors did not address this issue, but rather try to make a hand-waving argument. To be clear: that reviewer did not ask for a detailed 2D-crystallographic analysis or tomographic data (as was done in previous studies by others) – the point raised in this previous review simply was that micron-sized arrays of any kind (even if they are highly disordered) are something one could not possibly miss in either negative stain or cryoEM. Trying to get out of this by invoking higher dynamic turnover doesn't seem the best idea because dynamic effects are reduced to exactly zero once the sample is either stained or frozen. Point is: Fig 3 suggests that these patches are indeed micron sized and given current detectors used in (cryo)EM, these things would indeed be impossible to miss if assembled on GUVs (yet nothing sensible shows in the figures that are provided). The statement that FRAP studies show high dynamic turnover is also somewhat troubled because extended Fig 8 top row suggests that recovery on GUVs is quite slow (and apparently "0" in HeLa cells – bottom row). Another issue here is that the lateral packing interactions within the lattice model proposed in Fig 1 are quite extensive....this would suggest that these aggregates should show some sort of structure when assembled on the membrane surface. Unless the authors have supportive data, it would be better to not use cryoEM data at all because as is, they are quite confusing and detrimental. But then: cryoEM is, by far, the best direct approach to support the author's claims.

We agree that the previous cryoEM images could be those of multilamellar vesicles. It was difficult to obtain sufficiently clear EM images using GUVs due to several experimental limitations, including the micro-thickness of the ice due to the size of the GUVs, and thus we did not pursue the cryoEM further. Instead, we obtained data showing the striations formed by the GAS7 sheet on the lipid monolayer, using negatively-stained samples analyzed by transmission electron microscopy (revised Fig. 1c). We observed GAS7 on the monolayered membrane formed on the flat grid for EM. Then, we successfully observed similar striations of the GAS7 F-BAR domain fragment, GAS7b, and GAS7cb.

In HeLa cells, the dynamics of GAS7b and the GAS7 F-BAR domain fragment were different, in that the F-BAR fragment was quite stable while the GAS7b assembly underwent exchange. This indicated that the F-BAR fragment could assemble tightly, while GAS7b could not. However, our EM analysis in this revised manuscript indicated that GAS7b could also form a similar assembly to that of the F-BAR fragment *in vitro* (Figure 1c). We also performed a new FRAP analysis of both the F-BAR fragment and GAS7b on GUVs, by monitoring the GUV formation with Rhodamine-PE (Figure 3k, l, Figure S8a, b), which showed that the dynamics was slightly faster for GAS7b, as compared to that of the F-BAR fragment. This correlation between the *in vitro* results and in cells in the FRAP analysis appeared to indicate that the difference in the dynamics resulted from the characteristics of GAS7 itself. However, the recovery of GAS7b *in vitro* was slower than that in cells, and therefore, the GAS7b binding molecules on cells were also thought to be involved in the dynamics of GAS7b.

We cited the paper by Frost *et al.* (Cell, 2008, 132, 807-817) that showed the binding of F-BAR on flat membranes, which are known to deform into tubules. The assembly of GAS7 on a relatively flat membrane appeared to function for a flatter phagocytic membrane, and therefore GAS7 can be considered

to be a unique F-BAR domain that functions on large, flat membranes. We discussed this point (lines 337-343) as follows:

The side surface was previously shown to be used by the F-BAR domain of FBP17 to bind to the flat membrane, presumably before it deformed the membrane into a tubular shape by binding through the concave N-surface¹⁰. GAS7 did not induce significant membrane deformation when it bound to flat membranes under our conditions. This might arise from the membrane binding preferentially on the side surface of the F-BAR domain, as determined by the mutagenesis analysis (Fig. 3, Supplementary Fig. 5a)

Lipid substrates and assay conditions used: this is, perhaps, one of the biggest design flaws in the study.

- *GUVs were made from PS/PE/PC mixes (some of which have unknown acyl chain compositions) in the absence of any salt;*
- *liposome binding studies used a Folch-Fraction (which one?) from Avanti (no catalogue number is provided and checking their catalogue – Avanti doesn't offer a "Folch"-Fraction, just a brain lipid extract with 10%PS content...not too high) and buffers with rather high salt concentrations (200 and 300mM respectively);*
- *in vivo assays use, well natural and asymmetric bilayers.*

This lack in consistency seems to be a fatal flaw because the remodeling behavior of BAR-domain proteins is extremely sensitive to lipid-compositions, acyl chain lengths of lipids, salt, type of monovalent cation, counterions, and temperature – among a few parameters that are critical. Granted: doing these studies is challenging and any choice of lipids is, to some extent, arbitrary. That said: choices should be consistent across experimental approaches, and – more importantly – should be put into a "physiological perspective" at some point. None of that happens here and consequently, the in vitro data cannot reliably be correlated with any of the in vivo imaging studies, not can the in vitro studies be directly compared to each other. These limitations reduce the work to a pure phenomenological study that cannot support detailed mechanistic models.

First of all, we apologize for the mistake in the methods section. The brain Folch lipids are from Sigma, and have been widely used in BAR domain research, including analyses of the t-tubule biogenesis in muscle (Razzaq et al., 2001), endocytosis in cultured cells (Shimada et al., 2007), and possible functional implications in brain (Peter et al., 2004). We agree that the physiological relevance for the use of brain Folch lipids is weak here, but on the other hand, people would wonder whether the GAS7 F-BAR domain behaves similarly to the other BAR domain proteins.

In this manuscript, the physiological role of GAS7 is in phagocytosis by macrophages. A previous study showed that PS and PIP₃ can mimic the macrophage plasma membrane (Yeung et al., 2008). Accordingly, we made liposomes containing 20% PS and 5% PIP₃, as well as liposomes with 60% PS resembling the brain Folch lipids, as this is considered to be a similar percentage to that in the brain Folch lipids from Sigma (our TLC analysis). These liposomes are considered to have similar charge densities. These two reconstituted liposomes bound GAS7 similarly, at a physiological 150 mM NaCl concentration (Figure 2f). The choice of 200 mM for the initial liposome analysis was from the optimum condition for protein purification for the crystals, while the use of the 300 mM salt concentration was only for the high salt concentration experiments, to examine the salt concentration dependency of the GAS7 sheet formation on the membrane.

The GUV preparation is hindered by ions, including those derived from charged lipids including PS and PIP₃, and therefore, the efficient production of GUVs is often performed using a sucrose solution (Stein et al., 2017). When we made GUVs using a sucrose solution, we used a buffer with a similar osmolarity for proteins to avoid the tension application to the GUVs. In these GUV experiments, the experimental setting will result in a 75 mM NaCl concentration, because a 50% volume of the GUV solution was mixed with an equal volume of the protein buffer. We did not precisely describe the buffer composition for the GUV experiments, and we apologize for the misleading statement in the previous manuscript. Therefore, the *in vitro* experiments were mostly performed at salt concentrations of 75-200 mM, which are traditionally considered to be close to the physiological salt concentration in the cell.

We agree that the acyl chain variations of phospholipids affect the binding of the BAR domain proteins. The exploration of the acyl chain will require significant additional work on the liposomes with the phospholipids with defined acyl chains. These experiments exceed the scope of our paper, which is the characterization of the GAS7 F-BAR domain and its functions. Rather, the natural phospholipids contain a variety of acyl chains, and therefore the use of phospholipids from a natural source might more precisely reflect the physiological situation, because they are considered to be composed of phospholipids with various acyl chains.

Regarding the asymmetry, our liposomes were reconstituted by Folch fraction lipids or by purified phospholipids. Although the liposomes made from the Folch fraction had different lipids that do not reside on the cytoplasmic surface of the membrane, the liposomes made of the reconstituted lipids only contained phospholipids, and therefore, these liposomes were considered to form a similar membrane surface to the plasma membrane in cells.

Overall, we think the data for the GAS7 binding to the liposomes are physiologically relevant, and we would like to point out that GAS7 binding to the membrane occurs under all of the conditions we examined. Furthermore, the membrane binding and oligomerization *in vitro* correlated with the patch-like localization in HeLa cells, as well as the ability to restore the phagocytic defect in GAS7 knockout macrophages. Therefore, we think that our *in vitro* analysis conveys sufficient and valuable insights into the GAS7 assembly in cells.

R326A mutant: at some point the authors want to make up their mind how to utilize the results obtained with this mutant. As is, this is an incredibly messy and inconsistent story that – if anything – says that lateral assembly has little to nothing to do with phagocytosis. This mutant

- apparently binds lipids (though Fig 1 f/g are not consistent with the narrative given in the text or with what is shown in extended figure 4),*
- but doesn't aggregate (?) (extended figure 6, Fig 2j, Fig 3a, and the narrative are all over the place, not consistently backing up each other),*
- yet apparently can rescue some phagocytosis that in the author's model depends on lateral sheet formation.*

We agree that the results obtained with the R326A were difficult to interpret, because arginine has a positive charge that can interact with the membrane. Therefore, we omitted the R326A mutant data. In this revised manuscript, we introduced the D207R mutant, which is also thought to destabilize FFL2, because D207 contacted other residues of the F-BAR domain (Figure S3), indicating the importance of the physical configuration of FFL2 in the assembly of GAS7 on the membrane into a sheet. The D207R mutant had an apparently increased positive charge, and had similar membrane binding activity *in vitro*, by the liposome co-sedimentation assay (Figure 2i). Interestingly, the cross-linking of the D207R mutant was weaker than that of GAS7b *in vitro* (Figure 2m), suggesting that the binding to the membrane and the oligomerization are not the same phenomena. This D207R mutant could not assemble into patches in the HeLa cells and could not restore the phagocytic defects of the GAS7 knockout macrophages (Figure 3). Therefore, the membrane binding, as well as the oligomerization, of GAS7 was strongly indicated to be essential for phagocytosis.

Fig 2: for these data to match the narrative would require these experiments to be performed in solution AND in the presence of a lipid substrate.

For the revised Figure 2m showing the cross-linking experiments, we added the cross-linking analysis of GAS7b and the D207R mutant with liposomes, showing the difference in the cross-linking between GAS7b and the mutants in the presence of liposomes, as well as in their absence. The Δ FFL2 mutant did not bind to the membrane, and therefore we omitted the cross-linking analysis of the Δ FFL2 mutant.

Fig 3: panels e-g are inconsistent with panel dwhy is GAS7 concentrated on only one side in panel d while equally distributed all around in the other cases?

This confusion was from the different cutting planes of the confocal images relative to the phagocytic cup. We used a similar cutting plane in the revised Figure 3d to avoid confusion.

Less major:

Fig 1d: what is the weird feature at the top of the GUV? Seems that lateral assembly starts at a curved region?

The images used for the previous Fig. 1d were not appropriate, because the top curved part of the GUV was not always the start site of the GAS7 assembly. In this revised manuscript, we replaced the image of GUV binding to GAS7 with better images (Figure 1b). A small accumulation of GAS7 sometimes occurred,

as shown at the bottom of Figure 1b. However, the correlation between these accumulations and the GAS7 assembly is presently unclear, and we would like to analyze these phenomena in the future.

Figure 3h: GAS7cb mutant is not shown but referred to in the text.

We missed the reference to the previous Supplementary Fig. 8i. The data are now referred to in the revised text.

'The authors should make a clear statement early on that the work is really focused on only the Gas7b isoform. The crystallographic data shown for Gas7cb are not contributing anything significant (since all the additional domain densities are disordered and don't show ...in fact, since there are not differences, it appears unnecessary to show the crystallographic data for the full-length protein).'

We appreciate the kind advice from this reviewer. However, we would like to include the crystallographic data of GAS7cb, to show that the presence of the SH3 and WW domains did not alter the structure of the F-BAR domain and to show that the crystal packing of GAS7cb did not have any resemblance to the striations of GAS7 on the membrane. However, we omitted the figure showing the surface conservation and the electron density maps.

Refs:

Frost, A., Perera, R., Roux, A., Spasov, K., Destaing, O., Egelman, E.H., De Camilli, P., and Unger, V.M. (2008). Structural basis of membrane invagination by F-BAR domains. *Cell* 132, 807-817.

Peter, B.J., Kent, H.M., Mills, I.G., Vallis, Y., Butler, P.J., Evans, P.R., and McMahon, H.T. (2004). BAR domains as sensors of membrane curvature: the amphiphysin BAR structure. *Science (New York, NY)* 303, 495-499.

Razzaq, A., Robinson, I.M., McMahon, H.T., Skepper, J.N., Su, Y., Zelhof, A.C., Jackson, A.P., Gay, N.J., and O'Kane, C.J. (2001). Amphiphysin is necessary for organization of the excitation-contraction coupling machinery of muscles, but not for synaptic vesicle endocytosis in *Drosophila*. *Genes Dev* 15, 2967-2979.

Shimada, A., Niwa, H., Tsujita, K., Suetsugu, S., Nitta, K., Hanawa-Suetsugu, K., Akasaka, R., Nishino, Y., Toyama, M., Chen, L., et al. (2007). Curved EFC/F-BAR-domain dimers are joined end to end into a filament for membrane invagination in endocytosis. *Cell* 129, 761-772.

Stein, H., Spindler, S., Bonakdar, N., Wang, C., and Sandoghdar, V. (2017). Production of Isolated Giant Unilamellar Vesicles under High Salt Concentrations. *Front Physiol* 8, 63.

Yeung, T., Gilbert, G.E., Shi, J., Silvius, J., Kapus, A., and Grinstein, S. (2008). Membrane phosphatidyserine regulates surface charge and protein localization. *Science (New York, NY)* 319, 210-213.

Reviewers' comments:

Reviewer #5 (Remarks to the Author):

In the revised manuscript entitled "Phagocytosis is mediated by two-dimensional assemblies of the F-BAR protein GAS7" by Prof Suetsugu and colleagues, the authors have performed additional experiments and revised the manuscript according to reviewers' comments. Most of the concerns have been addressed in the revised version. However, there are a couple of issues:

In the previous version, formation of striations by GAS7 were examined using giant vesicles. In the revised manuscript, the giant vesicle was replaced by a lipid monolayer, as shown in Fig. 1c, 1d. However, a control should be included because the formation of a lipid monolayer is tricky, which might cause EM artifact as well. Furthermore, formation of striations by GAS7 mutants should be included in the Figure, particularly mutants with defects in membrane binding and protein oligomerization.

Protein oligomerization is enhanced in the presence of liposomes (Fig.2m), suggesting that lipid binding is important for protein oligomerization. The mutant D207R displays a similar membrane binding affinity with the wild type protein (Fig.2l). The authors indicated that this mutant exhibited reduced oligomer formation in vitro (Fig.2m). However, it appeared that oligomerization of D207R was similar to the wild type in the absence of liposomes while oligomerization of D207R was reduced in the presence of liposomes compared to the wild type GAS7b, suggesting that reduced oligomerization of D207R should relate to the membrane interaction. This is not consistent with the similar lipid binding affinity of D207R with the wild type protein.

Reviewers' comments:

Reviewer #5 (Remarks to the Author):

In the revised manuscript entitled "Phagocytosis is mediated by two-dimensional assemblies of the F-BAR protein GAS7" by Prof Suetsugu and colleagues, the authors have performed additional experiments and revised the manuscript according to reviewers' comments. Most of the concerns have been addressed in the revised version. However, there are a couple of issues:

In the previous version, formation of striations by GAS7 were examined using giant vesicles. In the revised manuscript, the giant vesicle was replaced by a lipid monolayer, as shown in Fig. 1c, 1d. However, a control should be included because the formation of a lipid monolayer is tricky, which might cause EM artifact as well. Furthermore, formation of striations by GAS7 mutants should be included in the Figure, particularly mutants with defects in membrane binding and protein oligomerization.

We would like to appreciate your constructive comments that significantly improved our manuscript. In this revised manuscript, we added the controls for EM, which are the images of the membrane alone and the protein alone. We also observed the D207R mutant on the membrane. The membrane binding of the D207R mutant was similar to that of GAS7b in liposome sedimentation assay but was not efficiently cross-linked in comparison with GAS7b, suggesting the difference in the oligomer formation on the monolayered membrane. Accordingly, the D207R mutant exhibited the reduction and the alteration in the formation of the striation on the membrane, forming somewhat discontinuous assembly on the membrane. Therefore, there indicated to be the difference between GAS7b and the D207R mutant in the oligomerization on the membrane (revised Figure 1c). Furthermore, the D207R mutant had one more positively charged arginine residue that would strengthen the binding to the membrane. Thus, the D207R mutation would compensate for the decrease in the binding to the membrane by the decrease in the oligomer formation, presumably explaining the reason for the similar binding of the D207R mutant to the membrane compared to that of GAS7b in the co-sedimentation assay. The K208A/K209A mutant had reduced membrane binding, and its striation was not apparent on the in vitro monolayered membrane presumably because of the reduced binding of the K208A/K209A mutant. For the clarity of the manuscript, we did not include the image of the K208A/K209A in the manuscript.

Protein oligomerization is enhanced in the presence of liposomes (Fig.2m), suggesting that lipid binding is important for protein oligomerization. The mutant D207R displays a similar membrane binding affinity with the wild type protein (Fig.2l). The authors indicated that this mutant exhibited reduced oligomer formation in vitro (Fig.2m). However, it appeared that oligomerization of D207R was similar to the wild type in the absence of liposomes while oligomerization of D207R was reduced in the presence of liposomes compared to the wild type GAS7b, suggesting that reduced oligomerization of D207R should relate to the membrane interaction. This is not consistent with the similar lipid binding affinity of D207R with the wild type protein.

The cross-linking happened between the reactive amino-acid residues and cross-linkers, and therefore the reduced cross-linking indicated that the relative positions of the amino-acid residues were altered by the D207R mutation on the membrane, which would suggest the orientation of GAS7b molecules on the membrane was altered by the mutation. As shown in the revised Figure 1c, the oligomerization of the D207R mutant on the membrane was reduced and different from that of GAS7b under electron microscopy. The D207R mutant had increased positively charged amino-acid residue, and was therefore supposed to compensate the possible decrease in the membrane binding due to the decrease in the oligomerization if we assume the oligomers should have higher membrane binding than the dimers. Therefore, we modified the statement that the D207R mutation as follows:

(page 8, line 208):

The D207R mutant exhibited fewer oligomer bands than GAS7b in the electrophoresis after the chemical-cross linking reaction in the presence of the liposomes, suggesting that the oligomerization of the D207R mutant was different from that of GAS7b (Fig. 2m, Supplementary Fig. 4h). Consistently, the striations of D207R mutant on the membrane were different and reduced from those of GAS7b (Fig. 1c). Because the increase of arginine would strengthen the binding to the membrane, the D207R mutation was thought to compensate for the reduction in the oligomerization that would contribute to the membrane binding, suggesting the reason of the membrane binding of the D207R mutant similar to that of GAS7b.

(page 9, line 260):

The D207R mutation abolished patch formation (Fig. 3a), which was thought to be consistent with the altered oligomer formation in vitro (Fig. 2m).